# HAS THE DEEP NEURAL NETWORK LEARNED THE STOCHASTIC PROCESS? AN EVALUATION VIEWPOINT

**Harshit Kumar, Beomseok Kang, Biswadeep Chakraborty & Saibal Mukhopadhyay**
School of Electrical and Computer Engineering
Georgia Institute of Technology
Atlanta, Georgia, USA
`{hkumar64,smukhopadhyay6}@gatech.edu`

## ABSTRACT

This paper presents the first systematic study of evaluating Deep Neural Networks (DNNs) designed to forecast the evolution of stochastic complex systems. We show that traditional evaluation methods like threshold-based classification metrics and error-based scoring rules assess a DNN's ability to replicate the observed ground truth but fail to measure the DNN's learning of the underlying stochastic process. To address this gap, we propose a new evaluation criterion called *Fidelity to Stochastic Process (F2SP)*, representing the DNN's ability to predict the system property *Statistic-GT*—the ground truth of the stochastic process—and introduce an evaluation metric that exclusively assesses F2SP. We formalize F2SP within a stochastic framework and establish criteria for validly measuring it. We formally show that Expected Calibration Error (ECE) satisfies the necessary condition for testing F2SP, unlike traditional evaluation methods. Empirical experiments on synthetic datasets, including wildfire, host-pathogen, and stock market models, demonstrate that ECE uniquely captures F2SP. We further extend our study to real-world wildfire data, highlighting the limitations of conventional evaluation and discuss the practical utility of incorporating F2SP into model assessment. This work offers a new perspective on evaluating DNNs modeling complex systems by emphasizing the importance of capturing the underlying stochastic process [1].

## 1 INTRODUCTION

Deep Neural Networks (DNNs) are increasingly employed to model complex systems and forecast their evolution across diverse fields, including infectious disease spread (Keshavamurthy et al., 2022; Ibrahim et al., 2021; Bomfim et al., 2020; Kuo & Fu, 2021), finance (Li et al., 2022), weather prediction (Scher & Messori, 2021; Bonavita & Laloyaux, 2020), geophysics (Yu & Ma, 2021; Tasistro-Hart et al., 2021), and wildfire prediction (Huot et al., 2022; Radke et al., 2019; Yang et al., 2021). These systems exhibit *complex global dynamics arising from relatively simple, localized, and stochastic interactions between participating elements*, which are often referred to as agents (Ladyman et al., 2013). Leveraging large-scale, multi-dimensional data—from satellite imagery to population records—DNNs aim to learn the agent interaction rules for long-horizon forecasting.

A fundamental characteristic of such physical systems is randomness, making their evolution inherently stochastic; thus, identical initial conditions can lead to multiple outcomes, forming a stochastic process. However, we typically observe only a single outcome of the system's evolution, a *system property* that we term the **Observed Ground Truth (Observed-GT)**. Current evaluation metrics primarily assess how closely a DNN's predictions match this Observed-GT, an *evaluation criteria* we call **Fidelity to Realization (F2R)**. This focus on F2R raises a critical question when a DNN fails to match the Observed-GT: is the mismatch due to inherent stochastic variability, or does it stem from exposure to a fundamentally different stochastic process the DNN has not modeled? Understanding this distinction is key: a DNN that captures the stochastic process but mismatches the Observed-GT may still be valuable, while failure to model the stochastic process entirely undermines its utility.

---

[1]Code—`https://github.com/harshitk11/evaluate_stochastic_process`

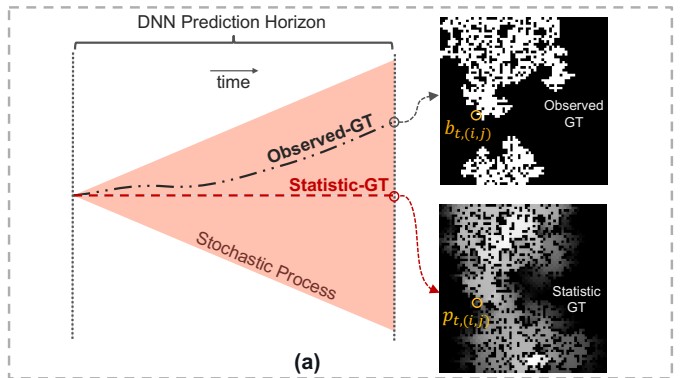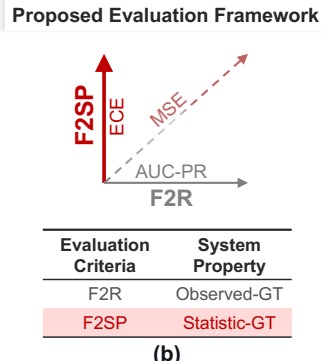

Figure 1: **(a)** The figure depicts the evolution of a stochastic process in a forest fire model. Starting from same initial conditions, diverse outcomes emerge over the prediction horizon, depicted by the shaded red region. The Observed-GT $\{b_{t,(i,j)}\}^{H \times W}$ represents one outcome on a $H \times W$ grid, while the **Statistic-GT** $\{p_{t,(i,j)}\}^{H \times W}$ shows the normalized frequency of target state occurrences, capturing the full stochastic process (§3). **(b)** This panel illustrates the proposed evaluation framework: **F2R** evaluates alignment with Observed-GT (using AUC-PR), **F2SP** tests alignment with Statistic-GT (using ECE), and MSE balances both criteria. The framework provides a unified approach for interpreting model performance in stochastic settings. See §H for a practical guide on the framework.

To address this problem, we formulate a new *system property* called the **Statistic Ground Truth (Statistic-GT)**. Statistic-GT captures the system's expected behavior across all possible outcomes from the same initial conditions. It provides a complete representation of the stochastic process and serves as a stable target for evaluation (Figure 1.a). With this new target, we propose an *evaluation criterion* called **Fidelity to Stochastic Process (F2SP)**, which measures how faithfully the DNN predicts the Statistic-GT. A key challenge arises from the fact that Statistic-GT is not directly observable in practical scenarios; only the Observed-GT is available, providing a partial glimpse of the complete stochastic process. *Thus, the challenge is testing F2SP using only the Observed-GT.*

To address this challenge, we establish that Expected Calibration Error (ECE) is a suitable evaluation metric that satisfies the necessary condition to test F2SP, whereas commonly used metrics like classification-based (e.g., Area Under the Precision Recall Curve, AUC-PR) and probabilistic error-based metrics (e.g., Mean Squared Error, MSE) fail to do so (Figure 1.b, §3). We conduct benchmark experiments on three synthetic complex systems—forest fire (Hargrove et al., 2000), host-pathogen (Sayama, 2013), and stock market (Wei et al., 2003) (§2)—demonstrating ECE's unique behavior across varying levels of stochasticity (§4). Finally, we evaluate on the real-world "Next Day Wildfire Spread" dataset (Huot et al., 2022), showcasing the limitations of F2R strategy and the practical utility of testing F2SP (§5). While ECE is traditionally used to estimate calibration error, our work discovers ECE's new utility for testing F2SP, broadening its role (§6). Our key contributions are:

- We identify critical limitations in current evaluation strategies focused solely on matching the Observed-GT, highlighting the need for a more robust approach in stochastic systems.

- We propose F2SP to assess DNN predictions against Statistic-GT, the system's expected behavior across all outcomes, ensuring stable evaluation in stochastic scenarios.

- We demonstrate, both formally and through benchmark experiments, that ECE uniquely satisfies the necessary condition for testing F2SP using only the Observed-GT.

- Beyond synthetic systems, we analyze real-world wildfire data, identifying instances where stochasticity disrupts traditional metrics and observing trends that align with our synthetic findings, reinforcing the practical applicability of our study.

By shifting the focus from replicating Observed-GT to capturing underlying stochastic dynamics, our work provides a novel perspective on evaluating DNNs that model complex systems. We recommend adopting the proposed evaluation framework (shown in Figure 1.b), which integrates ECE for assessing F2SP alongside F2R metrics, such as classification-based and proper scoring rules, to ensure a stochasticity-compatible evaluation strategy. This simple framework has significant implications for improving model evaluation, which this paper aims to explore and highlight.

## 2 BACKGROUND AND DATASET

This section formalizes the DNN prediction and evaluation framework (§2.1). It describes real-world challenges in observing the effects of stochasticity, motivating the use of synthetic benchmarks (§2.2). Finally, it covers synthetic systems used in this work and their stochastic simulation (§2.3).

### 2.1 FORMULATION OF DNN PREDICTION AND EVALUATION FOR COMPLEX SYSTEMS.

We model the evolution of a complex system on a grid of size $H \times W$, where each grid cell $(i, j)$ at time $t$ can occupy one of $m$ discrete states, denoted as $s_{t,(i,j)} \in \{s_1, s_2, \ldots, s_m\}$. We are particularly interested in tracking a specific state $s^*$. Let $b_{t,(i,j)} \in \{0, 1\}$ represent whether state $s^*$ is present (1) or absent (0) in cell $(i, j)$ at time $t$, collectively forming the grid-level ground truth $B_t = \{b_{t,(i,j)}\}^{H \times W}$. Additionally, $O_t \in (\mathbb{R}^n)^{H \times W}$ represents an $n$-dimensional vector of observational variables (e.g., environmental factors, agent attributes) for each of the $H \times W$ grid cells at time $t$. The DNN predicts the probability of state $s^*$ being present at each cell $(i, j)$, denoted by $\hat{p}_{t,(i,j)}$. These predictions form the joint conditional distribution $\hat{P}_t = \{\hat{p}_{t,(i,j)}\}^{H \times W}$, predicted $T$ timesteps into the future, using past states $B_{1:t-T}$ and observational variables $O_{1:t-T}$:

$$\hat{P}_t := \hat{P}(B_t \mid B_{1:t-T}, O_{1:t-T}) \tag{1}$$

After prediction, we use an evaluation metric $S(B_t, \hat{P}_t) : (\{0, 1\}^{H \times W} \times [0, 1]^{H \times W}) \to \mathbb{R}$ to measure prediction accuracy. Different choices of $S$ assess different system properties (see §3.4). In Appendix B, Table 3 serves as a reference for the notations used throughout the paper.

### 2.2 STOCHASTICITY IN COMPLEX SYSTEMS AND CHALLENGES IN STOCHASTIC MODELING

Stochasticity in complex systems stems from randomness in interactions driven by environmental factors, agent behaviors, and external interventions, adding noise to the "evolution rules" DNNs aim to learn. For example, the spread of wildfires is shaped by vegetation, terrain, weather, and human activity (Liu et al., 2021), while infectious disease dynamics depend on movement, social interactions, and interventions (Großmann et al., 2020). Small variations in these interactions can cause divergent outcomes, similar to the "butterfly effect" (Lorenz, 2000), complicating long-term forecasting.

Real-world data, providing only a single observed outcome, obscures the true implications of stochastic interactions in complex systems. To circumvent this limitation, we use simulation-based approaches like agent-based models (ABMs) (e.g., NetLogo (Tisue & Wilensky, 2004)), proven effective in capturing real-world complexities (Manson et al., 2012). See §C.1 for applications of ABMs in complex system modeling. In this study, ABMs generate a range of potential evolutions based on stochastic rules, allowing us to study emergent behaviors beyond real-world data constraints.

### 2.3 SYNTHETIC COMPLEX SYSTEMS AND THEIR STOCHASTIC SIMULATION

**Synthetic environments in this work** We use three synthetic environments—forest fire (Hargrove et al., 2000), host-pathogen (Sayama, 2013), and stock market models (Wei et al., 2003)—as examples

Table 1: Overview of the synthetic complex systems used in this study.

| Environment | Competitive | States | Interaction Rules | Description |
|---|---|---|---|---|
| **Forest Fire** | No | empty, patch, fire, ember $s^* = \{\text{fire}, \text{ember}\}$ | Fire spreads to neighboring patches with probability $p_{\text{ignite}}$, based on Rothermel's heat transfer model. | Simulates fire spread through land patches without opposing states. Agents spread fire without competition between states. |
| **Host Pathogen** | Yes | empty, dead, healthy, infected $s^* = \{\text{healthy}\}$ | Healthy agents are infected with probability $p_{\text{infect}}$, infected agents die with $p_{\text{dead}}$, and dead cells are cured by healthy cells with $p_{\text{cure}}$. | Models disease transmission and recovery, with competing healthy and infected states. |
| **Stock Market** | Yes | hold, sell, buy, inactive $s^* = \{\text{buy}\}$ | Investors buy, sell, or hold based on market sentiment and neighbor actions with probability $p_{\text{invest}}$. | Simulates competing investor actions (buy vs. sell), influencing market trends. |

of complex systems. While all three models consist of four discrete states $s_{t,(i,j)} \in \{s_1, s_2, s_3, s_4\}$, they have completely different interaction dynamics. Further, the Host-Pathogen and Stock Market models feature competing agents, reflecting socio-economic dynamics, unlike the natural processes in the Forest Fire Model. We assume homogeneous agents (no observational variables $O_{t,(i,j)}$) to focus solely on stochastic dynamics. See Table 1 for an overview and §D.1 for detailed descriptions.

**Agent-Based Simulation Framework.** We simulate on a $64 \times 64$ grid, where each cell $(i, j)$ at time $t$ represents an agent in state $s_{t,(i,j)}$. Simulations start with agents randomly assigned initial states $s_{0,(i,j)}$ based on a distribution $P(s_{0,(i,j)})$. Agents interact with their neighbors according to stochastic transition rules $P(s_{t+1,(i,j)} \mid \{s_{t,(k,l)}\}_{(k,l) \in \mathcal{M}_{(i,j)}})$, where $\mathcal{M}_{(i,j)}$ represents the agent's Moore neighborhood. These local interactions drive global behaviors, manifesting as complex, fractal-like patterns. The stochasticity in the interaction rules introduces randomness into the system's evolution, simulating the diverse behaviors. To control stochasticity, we use the **S-Level** parameter, with higher S-Levels introducing more randomness and S-Level 0 representing deterministic interactions.

**Role of S-Level and its realism.** The S-Level acts as a control knob for adjusting stochasticity. Its focus is not on realism, but on precise control of the stochastic process. This enables robust assessment of evaluation metrics' fidelity to the system property Statistic-GT across varying levels of randomness (see §3.2). In real-world datasets (§5), the concept of S-Level is more complex, as stochasticity can be represented using a distribution of S-Levels shaped by environmental and temporal dynamics. However, the fundamental property of Statistic-GT remains relevant.

## 3 EVALUATING A COMPLEX STOCHASTIC PROCESS

This section formalizes Statistic-GT as an evaluation target aligned with system stochasticity (§3.1, 3.2) and emphasizes its importance by highlighting the limitations of F2R strategy (§3.3). Finally, it provides theoretical insights showing why ECE uniquely tests F2SP using Observed-GT (§3.4).

### 3.1 SIMULATING A COMPLEX STOCHASTIC PROCESS

**Empirical Stochastic Process (ESP).** An ESP is represented through Monte Carlo (MC) simulations (1000 in our study), each starting from identical initial conditions and running until all state transitions cease. Identical initial conditions ensure variations in fire evolution arise solely from stochastic interactions. In the wildfire example, the forest configuration and fire seed location define the initial conditions, and the simulation runs until fuel depletion. Figure 2.a shows four distinct MC simulations (MC-1 to MC-4), illustrating different evolution pathways from the same starting point.

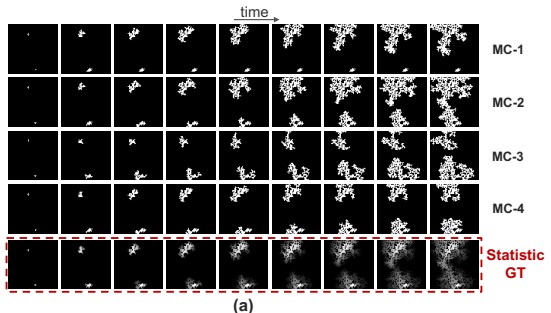 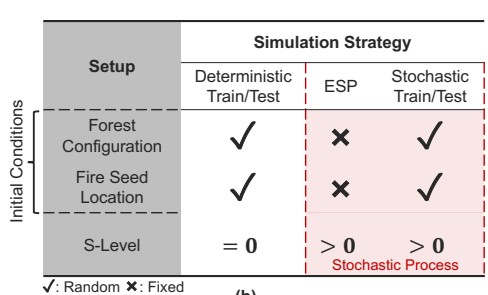

Figure 2: (a) The first four rows display four distinct MC simulations of forest-fire evolution from the same initial condition. The last row shows the Statistic-GT representing the evolution of the Stochastic Process (formally defined in §3.2)). For qualitative examples of MC simulations from other complex systems, please refer to §D.1.; (b) Table highlights the sources of randomness in the synthetic dataset across different simulation strategies used in this study. Deterministic processes randomize initial conditions but follow fixed fire evolution rules ($S$-Level $= 0$). Stochastic processes ($S$-Level $> 0$) allow multiple evolutionary paths: the ESP fixes initial conditions, while stochastic train/test setups (used in benchmark experiments, §4) randomize initial conditions across simulations.

**Random instances of deterministic vs. stochastic evolution.** Figure 2.b highlights the sources of randomness in the forest fire dataset: forest configuration, fire seed location, and fire evolution rules. In deterministic evolution, only forest configuration and fire seed location are randomized, while fire follows a single pathway, producing random instances of deterministic fire. In contrast, the ESP allows multiple evolutionary paths from the same initial conditions due to stochastic evolution. When all elements are randomized, we obtain random instances of stochastic fire evolution, used to train DNNs in §4.1. This synthetic dataset simulates a simplified version of real-world scenarios.

### 3.2 Formulating a Complex Stochastic Process

**Micro-Level Modeling.** At the micro-level, each grid cell $(i, j)$ at time $t$ is modeled using a binary random variable (RV) $M_{t,(i,j)}$, which captures the stochastic nature of the GT state $b_{t,(i,j)}$. $M_{t,(i,j)}$ follows a Bernoulli distribution with probability parameter $p_{t,(i,j)}$. This parameter is derived from the ESP by normalizing the frequency of $s^*$ across all MC simulations at each timestep.

**Statistic-GT.** While individual Micro RVs, $M_{t,(i,j)}$, represent independent probability parameters, the ensemble $P_t = \{p_{t,(i,j)}\}^{H \times W}$ captures the joint probability distribution across the entire grid as defined in Equation 1. These grid cells are spatially and temporally interdependent, forming the Statistic-GT. This emergent behavior of the micro-level statistic $p_{t,(i,j)}$ is a representation of the full stochastic process. The evolution of Statistic-GT is depicted in the fifth row of Figure 2.a. Typically, only a single instance from this stochastic process (one MC simulation) is observable in practice.

### 3.3 Why is Statistic-GT a Property of Interest: Limitations of F2R

In complex systems, evaluation metrics assess a DNN's performance by aggregating micro-level comparisons between the ground truth $b_{t,(i,j)}$ and predicted forecast $\hat{p}_{t,(i,j)}$ for each grid cell $(i, j)$. This aggregation yields a *macro-level* score that summarizes model performance across the grid, reflecting its ability to replicate the Observed-GT. However, in stochastic systems, reliance on a single observed instance of the system's evolution increases sensitivity to variability in the Observed-GT. This sensitivity creates challenges for the F2R strategy. We analyze this sensitivity by examining how variability in Observed-GT affects the stability of evaluation metrics, including classification-based metrics, MSE, and ECE, as detailed in §F.1, with key findings summarized in the main paper.

To formalize this analysis, we define the Macro RV $Z_t$, which represents the total number of grid cells in the target state $s^*$ at time $t$. This variable provides a grid-level summary of the system's evolution, analogous to the calculation of the evaluation metric. Next, we show that classification-based metrics are particularly sensitive to macro-variance ($Var[Z_t]$) because they exclusively test F2R. Higher $Var[Z_t]$ increases mismatches between Observed-GT and DNN predictions, leading to: (1) score degradation due to randomness and (2) metric fluctuations caused by GT variability, falsely indicating model instability. MSE and ECE are less sensitive to $Var[Z_t]$ because they do not exclusively test F2R. This observation underscores the instability of Observed-GT as a system property for stochastic systems, making F2R inadequate for evaluating models in highly stochastic scenarios.

In contrast, the Statistic-GT represents the system's expected behavior across all possible outcomes, offering a more stable evaluation target. Testing F2SP evaluates whether the model has learned the underlying stochastic process, ensuring robust performance assessment in highly stochastic systems.

### 3.4 How do we test Fidelity to Statistic-GT?

As stated in Equation 1, given a DNN's prediction $\hat{P}_t$ and the Observed-GT map $B_t$, the evaluation metric $S(B_t, \hat{P}_t)$ measures the fidelity of the DNN's predictions to a specific property of the system. Our focus is on the property Statistic-GT, denoted by $P_t$. *The key challenge is testing a DNN's fidelity to Statistic-GT using only Observed-GT, as Statistic-GT is not directly measurable.*

In our research, we discovered that ECE, an evaluation metric commonly used in Deep Learning and calibration error estimation, possesses the unique ability to test fidelity to Statistic-GT using only the Observed-GT. Next, we formally show how ECE achieves this and compare it to standard baseline metrics, that lack this capability. In §4, we empirically validate this discovery.

### 3.4.1 Expected Calibration Error Tests Fidelity to Statistic-GT

We formally show that ECE satisfies the necessary condition for evaluating a DNN's fidelity to the Statistic-GT. Consider a grid of size $H \times W$ at time $t$, where each cell $(i, j)$ is modeled as a Bernoulli random variable $M_{t,(i,j)} \sim \text{Bern}(p_{t,(i,j)})$, indicating the presence (1) or absence (0) of $s^*$. The DNN outputs predicted probabilities $\hat{p}_{t,(i,j)}$ for each cell, forming the set $\hat{P}_t = \{\hat{p}_{t,(i,j)}\}^{H \times W}$.

We group the predicted probabilities into $K$ bins $I_k = \left(\frac{k-1}{K}, \frac{k}{K}\right]$ and define $H_k = \{(i, j) \mid \hat{p}_{t,(i,j)} \in I_k\}$ as the set of indices in bin $I_k$. Let $b_{t,(i,j)} \in \{0, 1\}$ be the Observed-GT at cell $(i, j)$. For each bin $I_k$, the fraction of positives is: $\text{frac}(k) = \frac{1}{|H_k|} \sum_{(i,j) \in H_k} b_{t,(i,j)}$. A calibration curve plots $\text{frac}(k)$ against the representative predicted probability $\hat{p}_k$ (usually the bin midpoint). A perfectly calibrated model satisfies $\text{frac}(k) = \hat{p}_k$ for all bins. The Expected Calibration Error (ECE) (Naeini et al., 2015) summarizes calibration as: $\text{ECE} = \sum_{k=1}^{K} \frac{|H_k|}{N} |\text{frac}(k) - \hat{p}_k|$.

**ECE tests fidelity to Statistic-GT.** For a perfect predictor where $\hat{p}_{t,(i,j)} = p_{t,(i,j)}$ for all cells, we examine $\mathbb{E}[\text{frac}(k)]$. Since $b_{t,(i,j)}$ are realizations of $M_{t,(i,j)}$, we have $\mathbb{E}[b_{t,(i,j)}] = p_{t,(i,j)} = \hat{p}_{t,(i,j)}$. Therefore,

$$\mathbb{E}[\text{frac}(k)] = \frac{1}{|H_k|} \sum_{(i,j) \in H_k} \mathbb{E}[b_{t,(i,j)}] = \frac{1}{|H_k|} \sum_{(i,j) \in H_k} \hat{p}_{t,(i,j)} = \hat{p}_k.$$

This shows that, on average, $\text{frac}(k) = \hat{p}_k$ for a perfect predictor, implying zero ECE (see §F.2 for calibration curves). Intuitively, calculating $\text{frac}(k)$ marginalizes over data points in bin $I_k$, treating them as independent and ignoring dependencies among $p_{t,(i,j)}$ for all $(i, j) \in H_k$. *Thus, a low ECE satisfies the necessary condition for evaluating fidelity to Statistic-GT, but not the sufficient criterion.*

**Application of ECE in Deep Learning vs. Our Work.** While ECE is commonly used to assess the calibration of DNN predictions in tasks like image and text classification, our work uniquely investigates its properties for evaluating complex system forecasting. Unlike prior studies that simply *use* ECE for measuring DNN output calibration, we identify its unique suitability for assessing fidelity to a fundamental system property (Statistic-GT) that aligns with the stochastic nature of these systems. By addressing system randomness through F2SP, we position ECE as a complementary metric. Our study establishes perfect calibration as a necessary condition for testing F2SP, a fundamental insight overlooked in prior works, emphasizing that calibration should be central to the evaluation of such stochastic systems rather than secondary to discriminative performance (see §C.2 for details).

### 3.4.2 Baseline Evaluation Metrics

**Classification-Based Metrics (Ferri et al., 2009).** The prediction problem can be framed as a classification task, where the goal is to predict whether each grid cell will be in a specific state (e.g., presence of $s^*$). Metrics like Precision, Recall, F1-score, and AUC-ROC/PR are commonly used. These metrics, based on thresholding or ranking, are favored for distinguishing between Type I and Type II errors. They exclusively assess how well thresholded predictions match the Observed-GT.

**Proper Scoring Rules.** Scoring rules evaluate probability forecasts by ensuring the best score is achieved when the forecast matches the GT distribution (Gneiting & Raftery, 2007). Strictly proper scoring rules, such as MSE, Binary Cross-Entropy (BCE), and Continuous Ranked Probability Score (CRPS), are commonly used in forecasting. These rules promote sharpness by penalizing uncertain predictions and can be decomposed into calibration and sharpness components (Gneiting & Raftery, 2007; Ramos et al., 2018). For example, MSE can be decomposed using the Brier Score decomposition (Blattenberger & Lad, 1985):

$$\text{MSE} = \underbrace{\sum_{m=1}^{M} \frac{|B_m|}{N} (\text{frac}(B_m) - p_m)^2}_{\text{Calibration}} + \underbrace{\sum_{m=1}^{M} \frac{|B_m|}{N} \text{frac}(B_m)(1 - \text{frac}(B_m))}_{\text{Refinement}},$$

where the Refinement term promotes sharpness by penalizing uncertainty. This decomposition highlights a key distinction: scoring rules penalize uncertain predictions at the micro level, making them directly influenced by $Var[Z_t]$, whereas ECE focuses solely on calibration and remains unaffected by $Var[Z_t]$ in its convergence. We elaborate further on this distinction in §D.2.

# 4 BENCHMARK EXPERIMENTS

## 4.1 EXPERIMENTAL SETUP

We employ a single-layer ConvLSTM architecture for its simplicity (Shi et al., 2015), designing the ConvLSTM-CA variant to preserve spatial information essential for learning cellular automata interaction rules (§E.1). Our experiments span various S-Levels, with 1,000 simulations for each S-Level: 0, 5, 10, 15, and 20 for the forest fire model; 10, 15, and 20 for the host-pathogen model; and 5, 10, and 15 for the stock market model (see §D.1 for S-Level definitions). Simulations are initiated with randomized conditions and configurations, representing distinct random realizations of specific stochastic processes, with varying S-Levels indicating different process complexities.

Training and testing parameters include the observation period $t_{\text{obs}}$ and the prediction period $t_{\text{pred}}$. The forest fire simulations use $t_{\text{obs}} = 10$ frames and $t_{\text{pred}} = 50$ frames, whereas the host-pathogen and stock market models use $t_{\text{obs}} = 10$ and $t_{\text{pred}} = 20$ frames. For each S-Level, 700 simulations are used for training and 300 for testing. The DNN predicts the subsequent $t_{\text{pred}}$ frames after observing the first $t_{\text{obs}}$ frames. We employ BCE loss for model training.

While the main paper presents ConvLSTM-CA results for AUC-PR and MSE, extended evaluations across multiple DNN architectures are detailed in §F.3. This includes multi-layer, spatial bottleneck ConvLSTM variants (Shi et al., 2015) and Attentive Recurrent Neural Cellular Automata (AR-NCA) (Kang et al., 2024), which specializes in modeling locally interacting discrete dynamical systems.

## 4.2 TESTING ECE'S ABILITY TO ASSESS DNN'S FIDELITY TO STATISTIC-GT

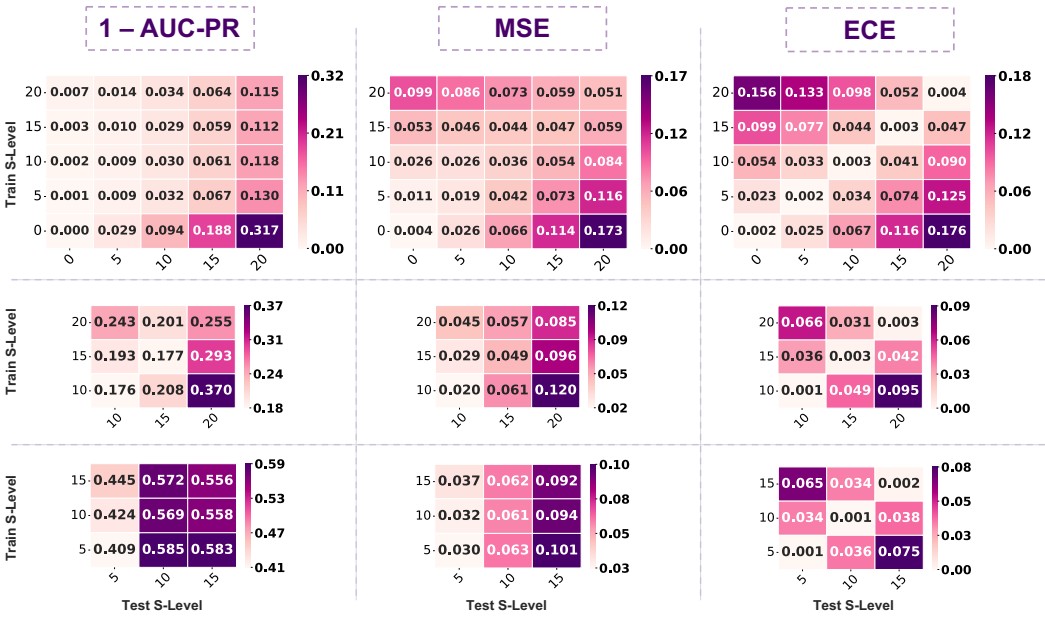

Figure 3: Performance of DNNs trained on one S-Level and tested on another, evaluated using three evaluation metrics: (a) $(1 - \text{AUC-PR}) \downarrow$, (b) MSE $\downarrow$, and (c) ECE $\downarrow$ across three complex systems. **Top Row:** Forest Fire, **Middle Row:** Host-Pathogen, **Bottom Row:** Stock Market. The $x$-axis (Test S-Level) and $y$-axis (Train S-Level) are consistent across all matrices. This figure highlights ECE's unique ability to evaluate whether the DNN has learned the correct stochastic process. While AUC-PR and MSE exhibit performance degradation as the difference between train and test S-Levels increases, ECE shows a distinct pattern. Specifically, for cases where the train and test S-Levels match, ECE indicates little to no performance degradation, underscoring its unique capability to test the F2SP evaluation criteria. Theoretical insights into these results are detailed in §3.4.1 and §3.4.2.

### 4.2.1 SENSITIVITY TO S-LEVEL: ECE VS. BASELINES.

In §E.2, we confirm that S-Level determines the Statistic-GT's appearance. We also verify that the DNN, trained with the self-supervised approach from §4.1, has learned to predict the Statistic-GT. Thus, an evaluation metric that measures a DNN's fidelity to Statistic-GT should exhibit sensitivity to the match between training and test S-Levels. To validate this, we create a matrix heatmap that evaluates DNNs (in §4.1), each trained at a specific S-Level, across test splits with different S-Levels.

Figure 3 presents the results, with each column representing a different evaluation metric and each row corresponding to a complex system. Within a matrix, rows represent evaluation scores for a DNN trained at a specific S-Level, and columns show scores for each test split at corresponding S-Levels. Only the ECE matrix shows clear diagonal behavior, indicating optimal scores when training and test S-Levels match. MSE shows partial diagonal trends, but its Refinement component weakens this pattern (§3.4.2), while AUC-PR shows no such trends. In §F.3, we confirm the generality of these findings across additional classification metrics (Precision, Recall, F1-Score, AUC-ROC), scoring rules (BCE, CRPS, Energy Score), and a spatial correlation-based metric.

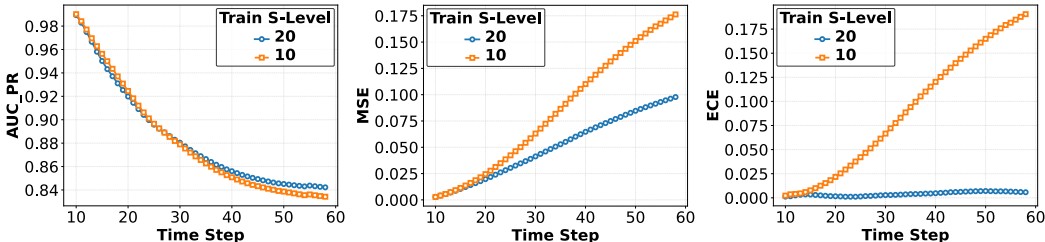

Figure 4: Two DNNs were trained on 700 forest fire simulations with different S-Levels—10 (orange, low stochasticity) and 20 (blue, high stochasticity)—and evaluated on 300 test simulations with S-Level 20. Evaluation metrics include (a) AUC-PR, (b) MSE, and (c) ECE, measured over an extended prediction horizon. AUC-PR shows similar trends for both models, failing to distinguish the stochastic mismatch. MSE exhibits a steeper decline for the mismatch case but also degrades for both models. In contrast, ECE remains low and stable for the DNN trained on S-Level 20, highlighting its unique ability to track alignment with the Statistic-GT. This behavior also highlights the potential stability in long-horizon predictions when tracking Statistic-GT, as it represents a single underlying property of the system, unlike Observed-GT, which varies across multiple possible outcomes.

### 4.2.2 LONG HORIZON BEHAVIOR OF ECE

As discussed in §3.3, evaluating fidelity to the Statistic-GT offers potential long-term stability due to the presence of a single Statistic-GT, unlike the wide range of Observed-GTs. We test whether this stability is reflected in the behavior of the evaluation metric. We hypothesize that ECE will remain low and stable over time when the DNN has learned the correct Statistic-GT, unlike traditional metrics which focus on Observed-GTs. Conversely, if there is a mismatch in the Statistic-GT, only ECE can exclusively capture this discrepancy, unlike baseline metrics.

Figure 4 shows two DNNs trained on different S-Levels (10 and 20) and tested on S-Level 20. For both DNNs, baselines like AUC-PR and MSE exhibit similar declining trends as prediction horizon increase, with MSE showing some distinction masked by system variance, and AUC-PR completely failing to differentiate between the models entirely. In contrast, ECE remains low and stable for the DNN trained on S-Level 20, indicating it has learned the correct Statistic-GT. Further exploration in §F.4 exhaustively confirms across all S-Levels that while baseline metrics decline with increasing S-Level and prediction horizon, ECE remains stable due to its alignment with the Statistic-GT.

**What makes ECE different from the baseline metrics?** Uncertain predictions at the micro level can collectively reveal macro-level properties like the Statistic-GT. ECE bins the predictions in $I_k$ and then compares the average predicted score $\hat{p}_k$ with the empirical fraction of positives frac$(k)$ in each bin. This property of ECE calculation allows for flexibility in micro-level mismatches while focusing on macro-level calibration, effectively capturing fidelity to the Statistic-GT (see §D.2).

## 5 CASE STUDY: REAL WORLD WILDFIRE PREDICTION

In this section, we apply our findings to the publicly available Next Day Wildfire Spread (NDWS) dataset, a large-scale multivariate dataset of U.S. wildfires aggregated via Google Earth Engine (Huot et al., 2022) (see G.1 for details on wildfire modeling). Unlike our synthetic experiments, where the S-Level provided precise control over stochasticity, the NDWS dataset involves unpredictable factors such as weather, vegetation, and human interventions, making it impossible to manipulate or quantify stochasticity. Given this limitation, we reassess prior work on the NDWS dataset, analyzing differences in the behavior of evaluation metrics like ECE, AUC-PR, and MSE, and interpret these observations in light of our benchmark experiments. We also explore how ECE can be integrated into the current evaluation protocol to help resolve conflicts in DNN ranking.

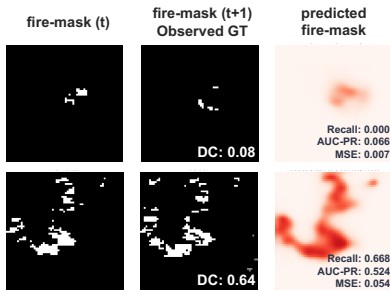

Figure 5: DNN takes as input a fire mask (left column) and 11 observational variables to predict the next-day fire mask (right column), compared to Observed-GT (middle column). F2R metrics indicate suboptimal performance.

Table 2: Evaluation Metrics for Conv-AE stratified by Dice Coefficient values measuring Fire Map Overlap (FMO) between $t^{th}$ and $t + 1^{th}$ day. (↑ means higher is better, ↓ means lower is better. Colors (normalized column-wise) indicate relative performance, with red representing worse scores and lighter shades representing better scores. ECE trends differently from F2R metrics.

| FMO (DC) | Sup. | Precision↑ | Recall↑ | AUC-PR↑ | MSE↓ | ECE↓ |
|---|---|---|---|---|---|---|
| 0.9-1.0 | 1 | 0.000 | 0.000 | 1.000 | 0.000 | 0.001 |
| 0.8-0.9 | 4 | 0.000 | 0.000 | 0.445 | 0.001 | 0.006 |
| 0.7-0.8 | 3 | 0.771 | 0.578 | 0.693 | 0.003 | 0.008 |
| 0.6-0.7 | 75 | 0.571 | 0.700 | 0.624 | 0.013 | 0.019 |
| 0.5-0.6 | 116 | 0.475 | 0.606 | 0.512 | 0.019 | 0.025 |
| 0.4-0.5 | 145 | 0.373 | 0.501 | 0.355 | 0.017 | 0.022 |
| 0.3-0.4 | 142 | 0.324 | 0.381 | 0.283 | 0.019 | 0.022 |
| 0.2-0.3 | 217 | 0.281 | 0.284 | 0.217 | 0.017 | 0.014 |
| 0.1-0.2 | 150 | 0.206 | 0.121 | 0.128 | 0.017 | 0.013 |
| 0.0-0.1 | 836 | 0.085 | 0.028 | 0.044 | 0.007 | 0.005 |
| Overall | 1689 | 0.346 | 0.311 | 0.247 | 0.012 | 0.012 |

**Forecasting problem.** The NDWS dataset compiles historical wildfire incidents into two-dimensional grids at a 1 km resolution, with 11 observational variables: elevation, wind direction and speed, temperature extremes, humidity, precipitation, drought index, vegetation, population density, and energy (§G.2) (Huot et al., 2022). The dataset includes 18,545 wildfire events, providing sequential snapshots of fire spread at times $t$ and $t+1$ day. A convolutional autoencoder-based DNN architecture, Conv-AE, is trained and tested on the data. The DNN takes as input a spatial map of the 11 variables along with the fire spread at time $t$ and outputs a binary map predicting fire spread at time $t + 1$.

**Observations in evaluation.** Huot et al. reported an AUC-PR score of 0.284 for Conv-AE on the test split. They noted that "while the metrics on the positive class seem low," qualitative visualizations showed that "fires are predicted," "predicted fires are roughly in the target location," and there is "good recognition of larger fires" (Huot et al., 2022). Similar observations are shown in Figure 5, displaying the fire mask at time $t$, the next-day fire mask at time $t + 1$ (Observed-GT), and the Conv-AE forecast. Despite the low scores suggesting poor performance, qualitative insights indicate otherwise. We hypothesize that the highly stochastic nature of wildfires limits the effectiveness of classification-based metrics, prompting a reevaluation of the DNN's performance.

**Revisiting evaluation of Conv-AE.** Huot et al. reported Precision, Recall, and AUC-PR for their model. We stratify these metrics by the Dice Coefficient (DC) between fire masks at times $t$ and $t + 1$, and report the scores in Table 2. Higher DC values indicate gradual fire progression, while lower values signify abrupt changes, often observed in smaller fires. In Figure 5, a DC of 0.08 represents a complete fire front shift, while a DC of 0.64 shows slower progression. Table 2 reveals distinct trends across the evaluation metrics. Classification-based metrics (Precision, Recall, and AUC-PR) degrade significantly as DC decreases, particularly in the lowest DC range (0.0–0.1). In contrast, MSE and ECE exhibit different behaviors. While MSE balances aspects of classification metrics and ECE, the latter shows notable improvement in performance as DC decreases. The calibration curve (§G.3) confirms that the DNN's predictions are well-calibrated at probability extremes, despite some overconfidence in the mid-range. Hence, *ECE behaves differently from classification-based metrics as DC varies*, and therefore measures a complementary aspect of performance. We attribute this behavior to ECE's exclusive focus on testing F2SP, as demonstrated in our benchmark experiments.

**Bridging metric conflicts.** In §G.4, we simulate selecting the optimal DNN from five DNNs trained on the NDWS dataset and observe rank conflicts between AUC-PR, MSE, and ECE. Such conflicts are common in model development, with no clear strategy for metric prioritization (Heaton et al., 2018). To address this, we introduce a cohesive evaluation framework shown in Figure 1.b. The framework places AUC-PR on the $y$-axis, aligning with F2R, and ECE on the $x$-axis, aligning with F2SP, reflecting their complementary roles. MSE sits between them, balancing both criteria. This framework ensures models are first validated for their understanding of stochastic dynamics (F2SP) before assessing accuracy for specific outcomes (F2R). A practical guide is provided in §H.

## 6    RELATED WORKS

Our interdisciplinary study bridges the application of deep learning to complex stochastic systems and their evaluation. While deep learning plays a crucial role in forecasting for fields like epidemiology, finance, weather, and geophysics, current evaluation strategies neglect the stochastic nature of these systems, focusing solely on F2R (see §C.3). This limitation extends to computer vision tasks—where many methods used in complex systems originate—such as stochastic video prediction (regression) and segmentation map forecasting (classification), where evaluation predominantly relies on the F2R strategy (see §C.4). Our work addresses this gap by introducing the evaluation criteria of testing F2SP. We also contribute to sensitivity analysis of evaluation metrics, extending prior studies on simple, low-dimensional setups with standard distributions (e.g., Normal, Exponential) to high-dimensional environments using stochasticity as the sensitivity variable (see §C.5).

## 7    CONCLUSION, LIMITATIONS, AND FUTURE WORK

We propose a new evaluation criterion to assess a DNN's ability to capture the stochastic interactions driving the evolution of complex systems. Through controlled experiments in a synthetic framework, we show that ECE uniquely evaluates this capability compared to classification-based metrics and proper scoring rules. Notably, ECE remains stable in long-horizon prediction performance, unlike other metrics. Applying these insights to a real-world wildfire dataset, we address the disconnect between positive qualitative assessments and negative performance scores and discuss an evaluation framework to mitigate rank conflicts between metrics. While fully understanding complex systems remains a challenge, they are crucial due to their societal impact. Our work highlights the need for improved DNN evaluation in stochastic scenarios, paving the way for more robust strategies.

**Limitations of ECE.** While ECE is effective for testing F2SP, it has limitations when compared to F2R evaluation strategy. Specifically, ECE has lower discriminative power than classification-based metrics in distinguishing between DNNs (§F.3). This limitation arises because calibration error lacks a refinement term to measure prediction sharpness, a critical aspect of F2R evaluation (§3.4.2). Further, ECE requires a sufficient number of samples for reliable convergence. In our experiments, which focused on multivariate problems, a test batch size of 10 grids (each $64 \times 64$) was sufficient for convergence (§F.5). Future work could explore improved calibration error estimators, such as those in (Gruber & Buettner, 2022), which offer stronger guarantees with smaller test sets. This approach could extend our study to uni-variate forecasting tasks, where sample size will be a key constraint.

**Applications beyond binary classifications and future work.** This paper focuses on binary and discrete prediction tasks, leaving extensions to regression tasks for future work. While calibration is intuitive for classification—where predicted probabilities should match observed frequencies—it becomes more complex for regression tasks (Kuleshov et al., 2018; Levi et al., 2022). Future work can explore adapting our findings to regression problems. While this work focuses on complex systems, it would be important to explore if the F2SP evaluation strategy benefits vision problems discussed in §C.4. Although such tasks are not traditionally classified as complex systems, their problem formulation and local pixel interactions in videos provide a compelling analogy.

A key limitation of the NDWS dataset is its restriction to next-day predictions, preventing long-term ECE tracking. Additionally, the absence of open-source complex system datasets—common in fields like Computer Vision and Natural Language Processing—hampers broader experimental validation. Future efforts should prioritize collecting and standardizing large-scale datasets for complex systems, with data spanning multiple time steps to enable long-term ECE monitoring. Assessing ECE stability and its potential benefits over time is a key avenue for future research.

ACKNOWLEDGMENTS

The authors thank Hemant Kumawat and Minah Lee for their insightful discussions, which significantly contributed to shaping the ideas presented in this paper. This work was supported in part by the Office of Naval Research under Grant N00014-20-1-2432. The views and conclusions contained in this document are those of the authors and should not be interpreted as representing the official policies, either expressed or implied, of the Office of Naval Research or the U.S. Government.

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

## A  OVERVIEW OF THE SUPPLEMENTARY MATERIAL

## B  NOTATIONS USED IN THE PAPER

Table 3: Summary of notations used in this study

| Symbol | Description |
|---|---|
| $H \times W$ | Grid size |
| $i, j$ | Location of a single cell within the grid |
| $s_{t,(i,j)}$ | State of a cell at time $t$ |
| $s^*$ | A specific state of interest |
| $b_{t,(i,j)}$ | $s^*$ present (1) or absent (0) at time $t$ in cell $(i,j)$ |
| $B_t$ | Observed-GT: System state across all cells at time $t$, $B_t = \{b_{t,(i,j)}\}^{H \times W}$ |
| $O_t$ | $n$-dimensional vector of observational variables across grid at time $t$, $O_t \in (\mathbb{R}^n)^{H \times W}$ |
| $\hat{p}_{t,(i,j)}$ | Predicted probability of state presence for cell $(i,j)$ at time $t$ |
| $\hat{P}_t$ | Predicted joint conditional distribution of system states |
| $p_{t,(i,j)}$ | Probability of state $s^*$ actually present in cell $(i,j)$ at time $t$ |
| $M_{t,(i,j)} \sim \text{Bern}(p_{t,(i,j)})$ | Micro random variable modeling $s^*$ presence/absence in cell $(i,j)$ at time $t$ with parameter $p_{t,(i,j)}$ |
| $P_t$ | Statistic-GT: Joint probability distribution of state $s^*$ across all cells at time $t$, $P_t = \{p_{t,(i,j)}\}^{H \times W}$ |
| $Z_t$ | Macro random variable representing the collective state of the system at time $t$ |
| $S(\cdot, \cdot)$ | Evaluation metric |

## C  LITERATURE SURVEY AND RELATED WORKS

### C.1  APPLICATION OF AGENT-BASED MODELS (ABMS) IN COMPLEX SYSTEMS.

ABMs, with each pixel acting as an individual agent, offer a natural way to simulate stochastic interactions. These models, defined by discrete space and time, and marked by local spatial interactions, align well with the physical nature of complex systems, e.g., (Gouveia Freire & Castro DaCamara, 2019). For example, in ABM forest fire models, each cell on fire is an agent capable of spreading the fire based on neighborhood interaction rules, leading to *emergent behaviors that mirror real-life fire propagation patterns* (Zinck & Grimm, 2008; Bak et al., 1987). In general, ABM has been applied to a wide range of complex systems in recent years. Key application areas include social-ecological systems, where ABMs have been used to study land-use changes, environmental adaptation, and biodiversity protection Schulze et al. (2017). In urban planning and traffic simulation, ABMs have modeled city growth, sustainability, and traffic flow dynamics Crooks et al. (2017). The COVID-19 pandemic has driven increased use of ABMs in public health and epidemiology for simulating disease spread and evaluating intervention strategies Hoertel et al. (2020). In economics and finance, ABMs have been employed to model market dynamics and assess economic policies Farmer & Foley (2009). Environmental management has benefited from ABM applications in forest management and climate change adaptation studies Belem & Saqalli (2018). These diverse applications demonstrate the versatility and power of ABM in capturing the intricacies of real-world complex systems.

### C.2  GENERAL USE OF ECE AS AN EVALUATION METRIC VS. OUR WORK

ECE is widely used to evaluate the calibration of DNNs in terms of confidence estimates and uncertainty quantification. Typically applied in static tasks such as image or text classification and object detection, ECE focuses on univariate predictions, where each test sample corresponds to a single outcome and prediction. In these contexts, studies have shown that a DNN's confidence is influenced by training and architectural factors. For instance, (Guo et al., 2017) observed that model capacity, batch normalization, and regularization techniques often lead to miscalibrated confidence estimates. Another significant challenge arises with out-of-distribution samples, where a DNN's lack of knowledge impacts the reliability of its confidence scores, prompting research into uncertainty quantification to address this limitation (Gal & Ghahramani, 2016).

Techniques such as temperature scaling (Guo et al., 2017), ensemble methods (Lakshminarayanan et al., 2017), Bayesian neural networks (Denker & LeCun, 1990), dropout (Gal & Ghahramani, 2016), and evidential deep learning (Sensoy et al., 2018), to name a few, aim to enhance calibration, with ECE serving as the standard metric for assessing calibration quality (Naeini et al., 2015). Additionally, ECE is widely employed in safety-critical domains like healthcare (Jiang et al., 2012) and cybersecurity (Kumar et al., 2021), where reliable confidence estimates are critical. Importantly, all these prior works focus on addressing the stochasticity in the model's predictive distribution $p(output|input)$. *This stochasticity is model-specific and reflects the uncertainty in DNN's predictions for a given input.*

However, in complex systems, stochasticity arises from the system's dynamics (e.g., agent interactions or environmental factors) which is fundamentally different from the model's uncertainty about its predictions. *ECE's conventional role does not naturally extend to capturing system properties like the Statistic-GT.* Beyond measuring calibration error for a DNN's output distribution, we demonstrate that ECE can evaluate F2SP by assessing predictions against the system property Statistic-GT using only Observed-GT. Our findings establish perfect calibration as a necessary condition for tracking fidelity to the Statistic-GT, a fundamental insight that significantly extends the utility of ECE.

Building on this expanded utility, our work is the first to discover ECE's unique utility in evaluating stochastic complex systems with high-dimensional, multivariate outputs, such as the 64×64 grids in our experiments. By addressing the system's inherent randomness through F2SP, rather than solely aligning with the Observed-GT as traditional F2R-focused strategies do, our findings position ECE as a complementary metric. While current model development primarily emphasizes discriminative performance through F2R strategy, calibration often takes a backseat (Naeini et al., 2015). For forecasting stochastic complex systems, however, calibration must take center stage, as it tests fidelity to fundamental system properties that inherently align with the stochastic nature of these systems.

### C.3 Complex Systems, Their Prevalence, and Evaluation Metrics

Deep learning (DL) is increasingly applied to complex systems, leveraging multi-dimensional data from sources like satellite imagery, population records, and the internet. These datasets enable DL models to uncover intricate patterns crucial for accurate forecasting. Recent applications include infectious disease prediction, where models integrate epidemiologic, geographic, and climatic factors to forecast outbreaks (Keshavamurthy et al., 2022). Ibrahim et al. used LSTM models for COVID-19 forecasts (Ibrahim et al., 2021), while Bomfim et al. applied mobility data to improve dengue transmission predictions (Bomfim et al., 2020). Kuo and Fu used county-level data to model COVID-19 infection rates (Kuo & Fu, 2021). In finance, Li et al. leveraged DL for asset price forecasting, capturing nonlinear dynamics (Li et al., 2022). In weather prediction, scoring rules like MSE and CRPS are commonly used (Scher & Messori, 2021; Bonavita & Laloyaux, 2020). DL has also been applied in geophysics, where models predict system states from geo-spatial data (Yu & Ma, 2021). Hart et al. used DL to forecast geomagnetic storms using satellite data, evaluating with RMSE, Pearson correlation, and calibration error to assess probabilistic accuracy (Tasistro-Hart et al., 2021). While most evaluations use probabilistic metrics (e.g., MSE, RMSE, MAE), classification metrics are applied in domains with well-defined outcomes, such as epidemic spread (Bomfim et al., 2020) and wildfire prediction (Huot et al., 2022). However, all approaches treat these problems deterministically through F2R formulation, and do not account for the inherent stochasticity of such complex systems.

### C.4 Evaluation Strategy for multivariate Forecasting in Computer Vision

In the computer vision community, evaluation strategies for forecasting tasks across both regression and classification problems predominantly focus on *matching the Observed-GT* (F2R). For instance, in stochastic video prediction—a regression task—DNNs generate a range of potential outcomes, and the score of the sample best matching the Observed-GT is reported (Oprea et al., 2022; Mathieu et al., 2016; Henaff et al., 2017). Similarly, for classification tasks such as predicting the evolution of segmentation maps, metrics like Intersection over Union and Average Precision (summarizing the precision-recall curve) are standard (Luc et al., 2017; 2018).

While this F2R-centric approach is sufficient for scenarios where stochasticity is sporadic (e.g., unexpected car turns in video prediction (Oprea et al., 2022)), it falls short for complex systems where stochasticity is inherent and consistent (Gallager, 2013). In such cases, our study emphasizes the importance of *matching the Statistic-GT* (F2SP) as a more robust evaluation strategy, particularly for high-risk, safety-critical applications. By providing stochastic interpretation and stochasticity-compatible evaluation methods, we aim to enhance the reliability of DNNs in such scenarios.

### C.5 Sensitivity Study of Evaluation Metrics

Our work aligns with broader research efforts on sensitivity analysis of evaluation metrics across different use cases, highlighting their strengths and weaknesses. Marcotte et al. (2023) explored how problem dimensionality and Monte Carlo approximation quality affect scoring rules' discriminative capability using synthetic univariate and multivariate distributions (e.g., Normal, Exponential). Pinson & Tastu (2013) and Alexander et al. (2024) compared Energy Score with other scoring rules under varying multivariate probabilistic forecast dependencies, modeled with Gaussian Distributions. Koochali et al. (2022) identified CRPS-Sum's limitations in distinguishing forecast quality in real-world time series with up to eight variables. Ferri et al. (2009) compared 18 classification-based metrics, analyzing their sensitivity to class threshold choice, ranking quality, calibration, and class distribution. Unlike these studies focusing on simple variables, we focus on characterizing evaluation metrics in *high-dimensional* stochastic complex systems, using stochasticity in interactions as the key sensitivity variable. Additionally, a different set of works examines the sensitivity of calibration error to various design choices, such as the binning process (Nixon et al., 2019; Roelofs et al., 2022). Unlike these studies, where the sensitivity variable is a property or design choice of the evaluation metric itself, our sensitivity variable is a parameter of the underlying complex system being evaluated.

Further, sensitivity of classification-based metrics have been studied in other contexts. Metrics like F1 score, precision, and recall are sensitive to evaluation protocols. In anomaly detection and security, factors like train-test splits, base rates, and decision thresholds impact these metrics (Fourure et al., 2021; Arp et al., 2022). In medical applications, Hicks et al. (2022) showed that interpretations of these metrics depend on class distribution and testing methods. In NLP, Yacouby & Axman (2020)

noted that these metrics evaluate only the top prediction, proposing alternatives. For geo-spatial grids similar to ours, Sofaer et al. (2019b) found AUC-PR more reliable than AUC-ROC due to its insensitivity to dataset imbalance. Contributing to this literature, we highlight a new key limitation: classification metrics become unreliable in highly stochastic systems.

# D  BACKGROUND

## D.1  DESCRIPTION OF THE SYNTHETIC COMPLEX ENVIRONMENTS

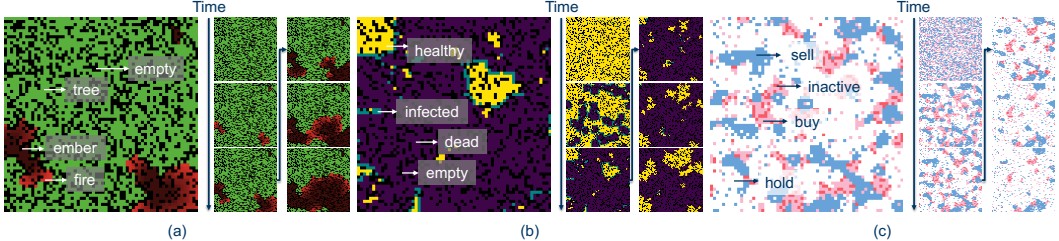

Figure 6: Figure shows the three different synthetic environments used in this work: (a) Forest Fire Model, (b) Host Pathogen Model, and (c) Stock Market Model. Each system includes four discrete states driven by different temporal dependencies.

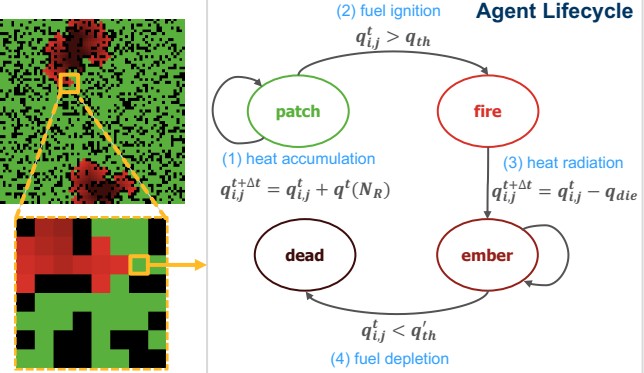

Figure 7: Snapshot of forest-fire evolution in NetLogo, using a 64x64 grid of locally interacting agents. Simulation is initialized by randomly assigning each agent in the grid a patch (green) or no-patch (black) value and different fire seed locations (3 in our case). Each patch agent goes through four stages, the flowchart of which is depicted in the right. The interaction between the agents leads to the emergent behavior of forest fire evolution. The agent's deterministic evolution rules are inspired from Rothermel's work on a heat transfer based model for wildfire evolution (Rothermel, 1972).

**Forest Fire Model.** Each grid cell $(i, j)$ represents an agent in one of four states $s_{t,(i,j)} \in \{\text{empty}, \text{patch}, \text{fire}, \text{ember}\}$ (denoted $s_1, s_2, s_3, s_4$). The target state consists of the burning states $s^* = \{\text{fire}, \text{ember}\}$. The simulation starts on a $64 \times 64$ grid, with each pixel initialized as a 'tree' or 'no-tree'. Agents have heat values $q_{(i,j)}$ crucial for the heat transfer in forest fire evolution. Fire seeds, placed at randomized (or fixed) locations, provide initial heat to agents. The initial condition is set as $q_{(i,j)} = I_{\text{seed}} \times q_{\text{threshold}}$ for seed locations $(i, j)$, and $q_{(i,j)} = 0$ otherwise, where $q_{\text{threshold}}$ is the ignition threshold and $I_{\text{seed}}$ amplifies seed heat values. A 'tree' agent accumulates heat from activated neighbors in its Moore neighborhood, in line with heat transfer mechanisms described by (Rothermel, 1972), following the equation:

$$q_{(i,j)}(t + \Delta t) = q_{(i,j)}(t) + \sum_{(k,l) \in N_R} \mathbf{1}_{(k,l)}(t) q_{(k,l)}(t)$$

The indicator function $\mathbf{1}_{(k,l)}(t)$ ensures only 'fire' state agents contribute to heat transfer. An agent's heat value $q_{(i,j)}$ exceeding $q_{\text{threshold}}$ triggers a state change from 'patch' to 'fire', and then to 'ember' in

the next time step. 'Ember' agents radiate heat at a rate of $q_{die}$ to adjacent non-fire patches, gradually losing heat until radiation ceases. This process ends when 'ember' agents darken, indicating $q_{(i,j)}$ falling below a certain threshold, thus terminating heat radiation and transitioning to the 'dead' state. The process is summarized in Figure 7.

To control the degree of stochasticity in the model, we introduce $p_{ignite}$, the probability of ignition after conditions are met (i.e., $q > q_{threshold}$). In deterministic scenarios, $p_{ignite} = 100\%$, ensuring ignition once the threshold is reached. We define the **S-Level** parameter as $(100\% - p_{ignite})$; higher S-Level values indicate more randomness in agent interactions, with S-Level 0 corresponding to deterministic interactions. Figure 2 shows four different Monte Carlo simulations of forest fire evolution.

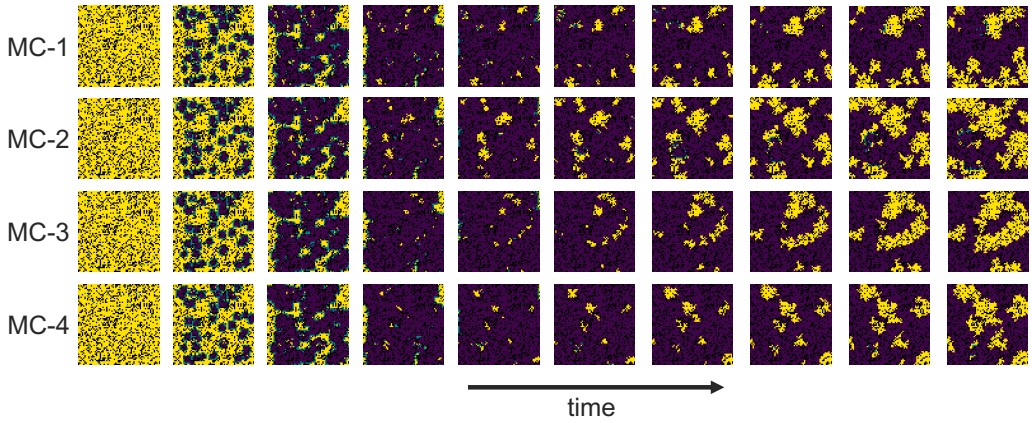

Figure 8: Figure shows four distinct Monte Carlo simulations of the evolution in the host pathogen system from the same initial condition.

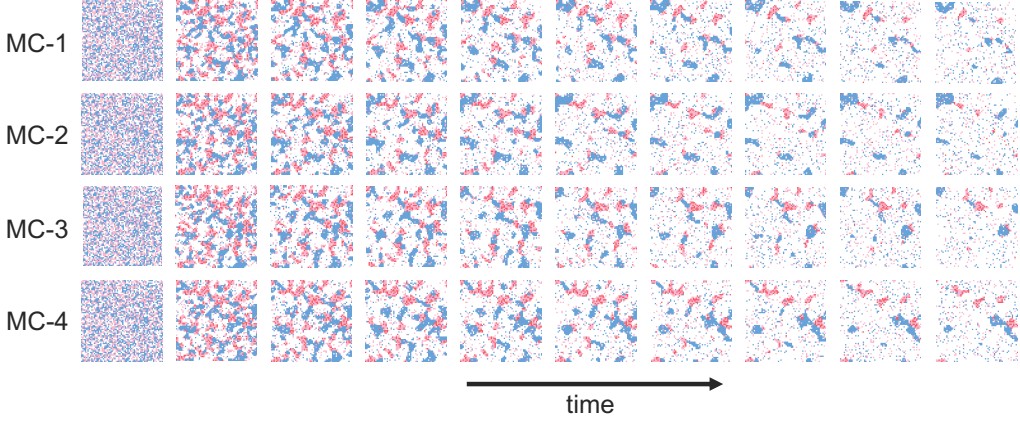

Figure 9: Figure shows four distinct Monte Carlo simulations of the evolution in the stock market system from the same initial condition.

**Host-Pathogen Model.** The Host-Pathogen model (Sayama, 2013) simulates virus-host interactions at the population level. Each grid cell $(i, j)$ can be in one of four states: empty, dead, healthy, or infected. We focus on the healthy state $s^* = $ healthy and define $M_{t,(i,j)} = \mathbb{I}(s_{t,(i,j)} = s^*)$. The system evolves as follows: 1) Create a $64 \times 64$ grid initialized with randomly distributed infected (1%), healthy (75%), and empty (24%) cells. 2) Each dead cell is cured by neighboring healthy cells with a probability $p_{cure} = 0.15$. 3) Each healthy cell is infected by neighboring infected cells with a probability $p_{infect} = 0.85$. 4) Infected cells transition to dead cells in the next timestep. 5) Repeat steps 2 to 4. This cycle models pathogen spread and recovery dynamics based on probabilistic transition rules. To control the appearance of Statistic-GT, we used $p_{cure}$ as the S-Level parameter, varying it between 10%, 15%, and 20%. Figure 8 shows four different Monte Carlo simulations of system evolution from the same initial condition.

**Stock Market Model.** The Stock Market model (Wei et al., 2003) uses cellular automata to simulate investor behavior influenced by neighboring investors. Each cell $(i, j)$ represents an investor in one of four states: hold, sell, buy, or inactive. We focus on the buying state $s^* = $ buy, defining $M_{t,(i,j)} = \mathbb{I}(s_{t,(i,j)} = s^*)$. The system evolves as follows: 1) Create a $64 \times 64$ grid with randomly assigned states (buy, sell, hold, or inactive). 2) Each cell transitions stochastically based on the dominant state of neighboring cells, following a transition matrix parameterized by market status $M = 0.05$ (positive market sentiment) and investment probability $p_{\text{invest}} = 0.95$ (Wei et al., 2003). 3) Cells that have been in the buying state for two consecutive timesteps become inactive. 4) Inactive cells from the previous sequence transition back to buying. 5) Repeat steps 2 to 4. To control the appearance of Statistic-GT, we introduced the S-Level parameter as $100 - p_{\text{invest}}$, resulting in S-Levels of 5%, 10%, and 15%. This models investor dynamics influenced by market conditions and peer interactions. Figure 9 shows four different Monte Carlo simulations of system evolution.

### D.2 PROPER SCORING RULES PENALIZE UNCERTAIN PREDICTIONS AT MICRO SCALE

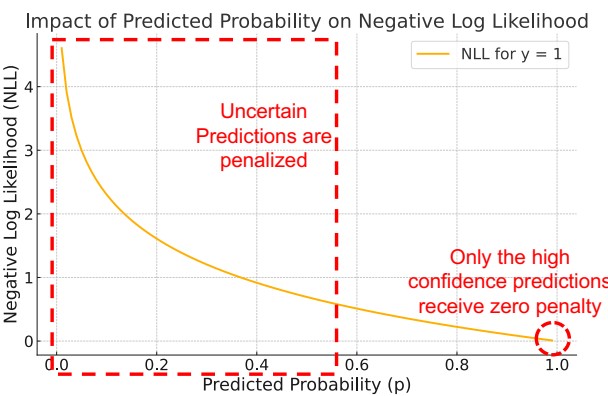

Figure 10: NLL for positive outcomes as a function of predicted probability in a binary classification problem. NLL penalizes uncertain predictions at the micro-level detail.

To clarify why proper scoring rules cannot test fidelity to the Statistic-GT, consider evaluating a single cell within the $64 \times 64$ grid using Binary Cross-Entropy (BCE), which computes the Negative Log Likelihood (NLL) of the true label $y \in \{0, 1\}$ given the predicted probability $p$. NLL is an optimal metric for evaluating probabilistic forecasts (Neyman & Pearson, 1933), calculated as:

$$\text{BCE}(y, p) = -[y \log(p) + (1 - y) \log(1 - p)].$$

As shown in Figure 10, NLL penalizes uncertain predictions—those with predicted probabilities away from 0 or 1—even if they are accurate in expectation. Only high-confidence predictions receive minimal penalty. Since proper scoring rules promote sharpness, they penalize uncertain predictions at the micro-scale when they do not match the Observed-GT.

In contrast, the *binning process* in ECE allows flexibility in micro-scale mismatches as long as the overall predictions are calibrated. This enables ECE to test fidelity to macro properties of the stochastic complex system, such as the Statistic-GT.

## E DNN CHARACTERIZATION

### E.1 DESIGN RATIONALE BEHIND CONVLSTM-CA AND CONV-CA

We choose ConvLSTM for its ability to model spatiotemporal systems, aligning well with the characteristics of the complex systems. Our modified version (see Figure 11.(a)), convLSTM-CA, places a ConvLSTM cell with a $3 \times 3$ kernel between an Encoder (with a $3 \times 3$ kernel) and a Decoder (with a $1 \times 1$ kernel). The Encoder takes an RGB image and transforms it into a latent tensor during a 10-timestep observation window. This tensor is then processed by the ConvLSTM cell, *maintaining its spatial dimensions*, before the Decoder produces a burnt map grid of softmax probabilities.

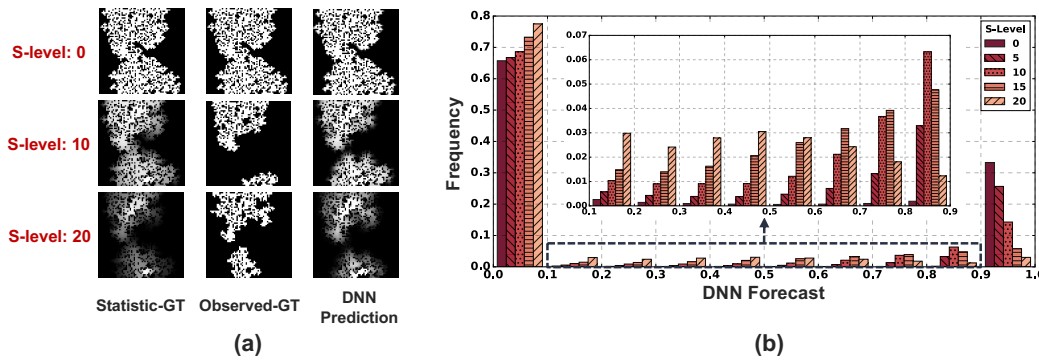

Figure 11: (a) Auto-regressive training of the DNN, (b) Softmax probability map from a convLSTM with a bottleneck, and (c) Softmax probability map from convLSTM-CA with no bottleneck.

This design choice of preserving spatial dimensions is crucial for modeling the cellular automata application. Reducing the spatial dimensions of the latent tensor is a popular design choice in video prediction models such as recurrent neural network-based (Wang et al., 2017; 2018) and simple CNN-based models (Gao et al., 2022). This reduction is often employed to decrease computational complexity and to capture essential spatial features while discarding less informative details. However, we observe that adding a bottleneck causes individual pixels (agents) to lose their identity. Maintaining these dimensions helps preserve each agent's identity, a critical factor for developing a DNN that minimizes mis-calibrated forecasts arising from limited model capacity. For instance, as seen in Figure 11. (b), using a compressed latent dimension results in a forecast cloud around predictions for deterministic fire evolution scenarios, which does not accurately reflect the system's true evolution. In contrast, as shown in Figure 11.(c) an uncompressed latent space yields predictions without a forecast cloud, aligning closely with the deterministic system's true evolutionary rules. Conv-CA used in Section G.4, is a version of Conv-AE (Huot et al., 2022) with the spatial bottleneck eliminated.

## E.2    IMPACT OF CHANGING S-LEVEL ON STATISTIC-GT AND DNN PREDICTIONS

Figure 12: (a) Visualizations of forest-fire snapshots for different S-Levels (one per row), showing Statistic-GT, Observed-GT, and corresponding DNN predictions (raw forecast values). DNN predictions closely resemble Statistic-GT. (b) Histograms showing the frequency of DNN forecast values for different S-Levels. The DNN's predictions become less confident as S-Level increases.

In this section, we analyze the impact of changing S-Level on Statistic-GT and DNN's raw predictions, and further demonstrate that the DNN learns to predict the Statistic-GT in highly stochastic scenarios. For a given initial condition (agent configuration, seed location), the S-Level uniquely determines the macro-scale pattern observed in the Statistic-GT. This is evident in the first column of Figure 12.a, where all three S-Level rows share the same initial condition but differ in S-Level. As S-Level increases, the cloudiness in the macro pattern also increases, confirming that S-Level uniquely determines the macro pattern in the Statistic-GT for a given initial condition.

In the second and third columns of Figure 12.a, we present the Observed-GT for a given MC sample and the corresponding DNN prediction on that sample. Higher S-Levels correlate with greater cloudiness in the DNN output. While the DNN's predictions begin to resemble the Statistic-GT, the corresponding Observed-GT can vary significantly and may not match the DNN's prediction.

To further investigate the alignment between the DNN's predictions and the Statistic-GT, we plot the mean Statistic-GT against the corresponding mean DNN forecast in Figure 13. The strong correlation suggests that the DNN is learning to predict the Statistic-GT (see qualitatives in Figure 14). This capability is linked to the use of Binary Cross-Entropy (BCE) loss—a proper scoring rule—which promotes calibrated forecasts (Gneiting & Raftery, 2007).

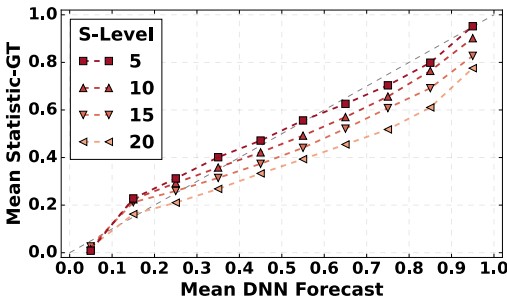

Figure 13: Correlation between Statistic-GT and DNN forecasts, indicating that the DNN is predicting the system property Statistic-GT.

As the DNN learns the Statistic-GT, it begins making more uncertain predictions. Figure 12.b illustrates histograms of DNN predictions across 1000 simulations for each S-Level. In deterministic settings, predictions are primarily binary (0 or 1), but at S-Level 20, predictions span the entire 0-1 range, indicating increased predictive uncertainty. However, these uncertain predictions still convey valuable information and contribute to a larger pattern that emerges at the macro scale.

Due to the stochastic evolution, it is inherently unachievable for the DNN to replicate the Observed-GT; instead, it learns to predict the Statistic-GT. If a DNN fails to predict the Statistic-GT, it indicates that it has learned a different stochastic process, making it fundamentally unsuitable for that scenario.

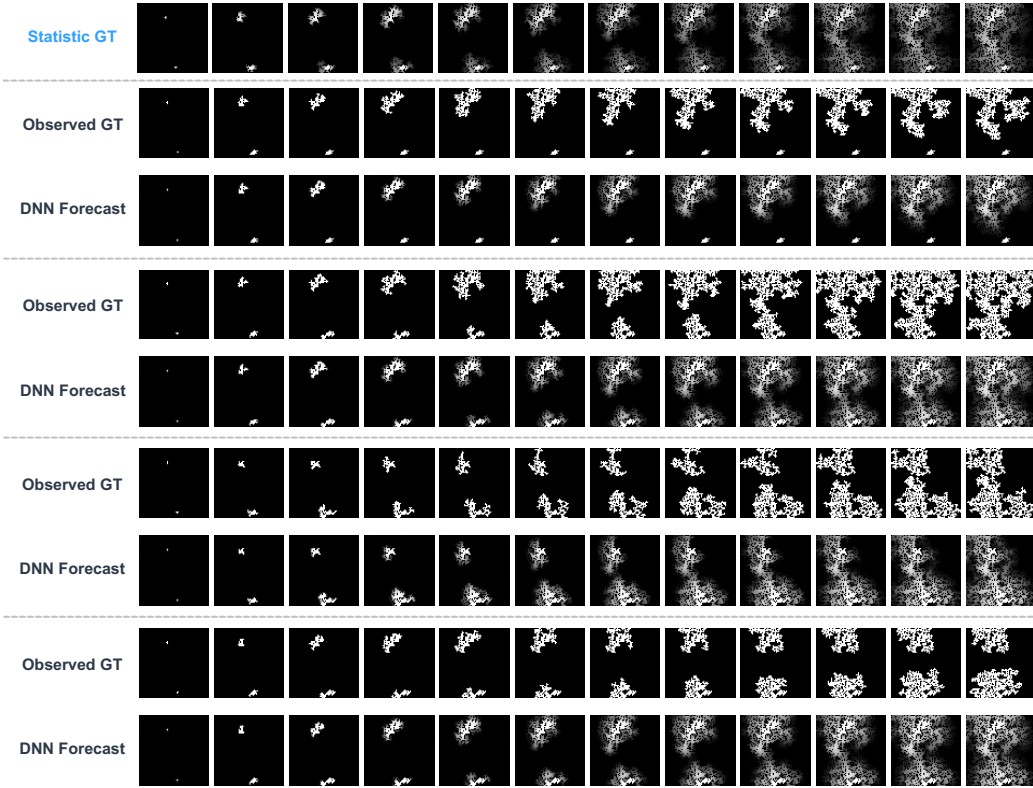

Figure 14: Qualitative visualizations of the DNN's forecasts across four different Monte Carlo simulations for the forest fire dataset. Frames are shown every five time steps. The DNN observes the first 10 time steps (first 2 frames) and predicts the next 50 time steps (last 10 frames). We can observe the difference between the DNN's forecasts and the Statistic-GT, which arises because of the determinism that is injected into the DNN's predictions due to its observation of the first 10 frames of the fire evolution. However, the DNN's predictions closely resemble the evolution of Statistic-GT.

# F    ADDITIONAL RESULTS

## F.1    IMPACT OF MACRO-VARIANCE ON FIDELITY TO REALIZATION STRATEGIES

In this section, we show that classification-based metrics like Precision, Recall, and AUC-PR, which test F2R, are highly sensitive to macro variance, exposing the limitations of Observed-GT as an evaluation target. The section motivates the use of Statistic-GT as a more stable alternative for assessing model performance in stochastic systems.

**Macro Random Variable.** To formalize the system-level modeling of stochastic fire evolution, we leverage insights from statistical mechanics (Eastman, 2015). At the microscopic level, individual agent behavior is captured through the Micro RV, $M_{t,(i,j)}$. At the macroscopic level, the system behavior is represented by the Macro RV, $Z_t$. Formally, $Z_t$ is defined as:

$$Z_t = \sum_{i=1}^{H} \sum_{j=1}^{W} M_{t,(i,j)}$$

Applying the Central Limit Theorem for a large number of Micro RVs ($\approx 10^3$), $Z_t$ can be modeled using a Normal distribution, characterized by its mean $E[Z_t]$ and variance $Var[Z_t]$. Sampling from $Z_t$ provides a macrostate value, representing the aggregate state of all agents. It should be noted that multiple microstates can correspond to the same macrostate value. Overall, $Var[Z_t]$ captures the system's tendency to explore diverse macrostates, with higher variance indicating greater overall unpredictability. The parameters of the Macro RV are extracted from the ESP by recording the number of unburnt trees (macrostate) at each time step. Using 1000 MC simulations, we generate a distribution at each time step to compute $E[Z_t]$ and $Var[Z_t]$.

### F.1.1    CHARACTERIZING THE S-LEVEL TEST CASES.

Figure 15.(a) shows the Macro RV $Z_t$ over time, with colors from red (S-Level=0) to black (S-Level=50) in increments of 5. The central line and shaded areas represent $E[Z_t]$ and $Var[Z_t]$. Figure 15.(b) highlights $Var[Z_t]$ over time for select S-Levels (0, 5, 10, 15, 20) used in our benchmarks. Key observations: At S-Level 0 (deterministic), the absence of $Var[Z_t]$ reflects a singular evolutionary path. Generally, $Var[Z_t]$ increases with S-Level and time, peaking at S-Level 20, indicating maximum variance and chaotic behavior typical of real-world systems (Zinck & Grimm, 2008). Beyond S-Level 20, fire dies out, reducing active pixels and $Var[Z_t]$. This setup allows us to test evaluation metrics under varying stochasticity, with S-Level 0 being deterministic and S-Level 20 representing peak randomness.

### F.1.2    SENSITIVITY OF F2R EVALUATION STRATEGY TO THE SYSTEM MACRO-VARIANCE

**Methodology.** For this experiment, we perform inference on the DNN trained with S-Level 20. The test dataset comprises 1000 MC simulations of the S-Level 20 ESP forest fire test case, all

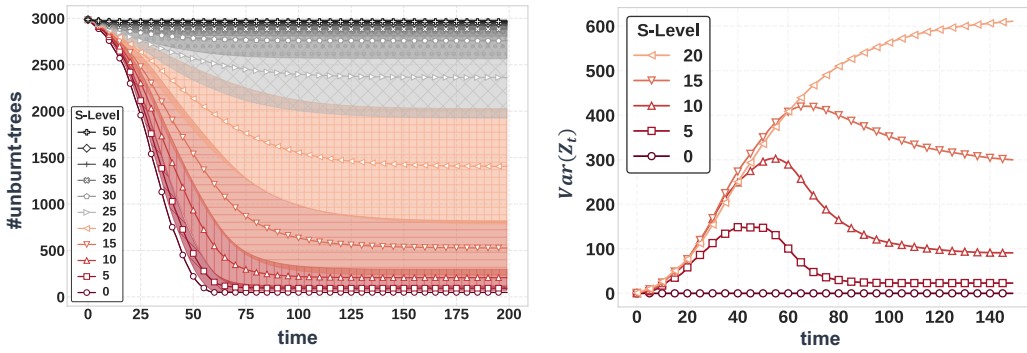

Figure 15: Characterizes the Macro RV $Z_t$: (left) ESP representing $Z_t$ across different S-Levels: mean is the center line (Statistic-GT) and variance is the shaded region; (right) $Var[Z_t]$ over time for selected S-Level test cases used in this study. S-Level 20 shows chaos (peak variance).

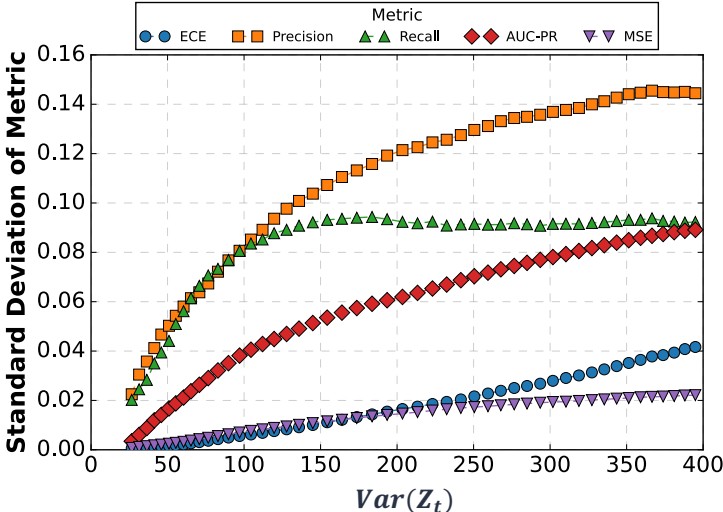

Figure 16: SD of evaluation metrics versus $Var[Z_t]$ at S-Level 20. Higher SD indicates increased sensitivity to $Var[Z_t]$. Classification-based metrics (Precision, Recall, AUC-PR) show greater sensitivity in highly stochastic environments, highlighting their unreliability in these settings. In contrast, MSE and ECE exhibit reduced sensitivity as they do not exclusively test F2R.

with identical initial conditions. We calculate the evaluation metric independently for each timestep of each MC simulation. Each simulation at a given timestep $t$ has a specific $Var[Z_t]$, reflecting macro-variance at that moment. We measure the Standard Deviation (SD) of the evaluation metric across the 1000 MC simulations against $Var[Z_t]$ to assess the metric's sensitivity to macro-variance. Since all simulations originate from the same stochastic process, an ideal metric that is faithful to the stochastic process, should demonstrate minimal sensitivity to individual realizations of the ESP.

**Results.** Figure 16 shows the SD of evaluation metrics against $Var[Z_t]$. The significant increase in SD for classification-based metrics (Precision, Recall, AUC-PR) highlights their heightened sensitivity to macro-variance and reduced reliability. Precision and Recall suffer high variance in stochastic settings due to their reliance on thresholding, which can produce predictive outcomes misaligned with the Observed-GT. This issue stems from the incompatibility of "thresholding" with stochastic systems, extending beyond the common critique of arbitrary threshold selection (Fourure et al., 2021). Although AUC-PR is less sensitive than other threshold-based metrics, it encounters challenges from its integration of False Positives (FP) through Precision, where the traditional concept of FP loses relevance because any non-zero forecast by the DNN renders both outcomes plausible.

In contrast, MSE and ECE demonstrate lower sensitivity to macro-variance as neither exclusively focuses on testing F2R. While both metrics exhibit similar trends in SD vs. $Var[Z_t]$, they converge to different steady-state values. Specifically, the steady-state value of MSE is influenced by $Var[Z_t]$, making it a function of this variance, whereas ECE's steady-state value remains unaffected by $Var[Z_t]$ in its convergence (see Figure 17). This theoretical distinction is further explored in §3.4.2.

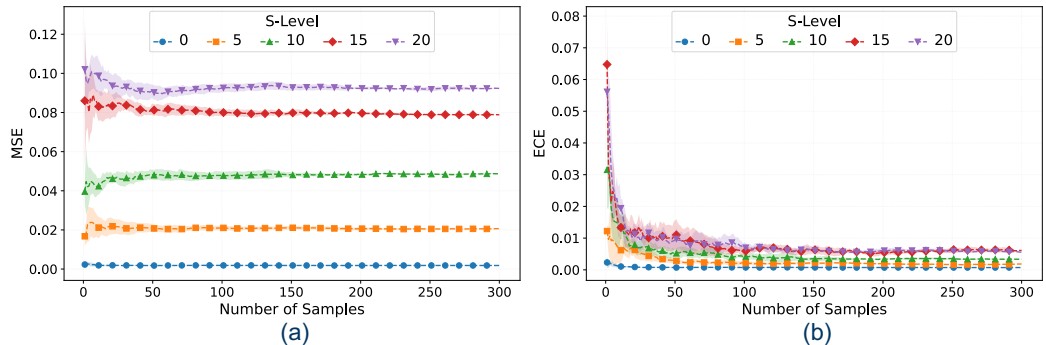

Figure 17: Testing the asymptotic convergence of MSE and ECE. The DNN is trained and tested on the same S-Level. Each sample is a grid of $64 \times 64$ predictions at t=59 in the test split of the synthetic forest dataset. As the number of samples increase in the calculation of MSE and ECE, they both converge. The MSE converges to a value that incorporates the $Var(Z_t)$, while the ECE converges to a low value as it excludes the impact of $Var(Z_t)$.

## F.2 Calibration Curve of the perfect predictor predicting Statistic-GT

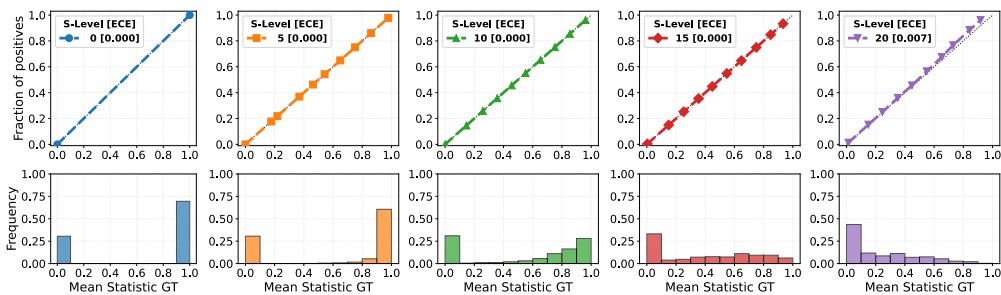

Figure 18: Calibration of a perfect predictor (predicting Statistic-GT) for different ESP test cases.

### F.3 GENERALIZABILITY OF ECE'S DIAGONAL BEHAVIOR

#### F.3.1 OTHER DNN ARCHITECTURES

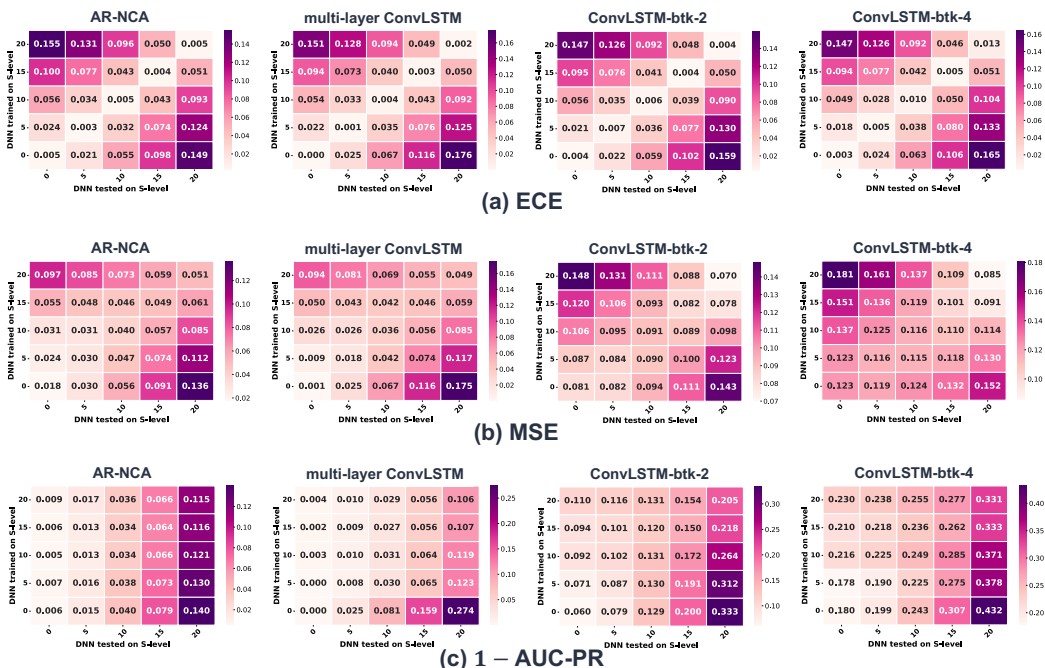

Figure 19: Evaluation metric scores for four different DNNs, from strongest (leftmost) to weakest (rightmost) using evaluation metrics (a) ECE, (b) MSE, (c) AUC-PR, and (d) Recall. While MSE, AUC-PR, Recall depict marked degradation in performance, ECE remains fairly stable, though slight increase is observed. In general, ECE shows lower variation on the synthetic dataset, indicating that it has a lower discriminating power.

Details about the DNN Architectures: (1) AR-NCA (Kang et al., 2024): AR-NCA involves a recurrent cellular attention module that couples LSTM and cellular self-attention, (2) multi-layer ConvLSTM (num_layers=2), (3) convLSTM-btk-2: convLSTM with a spatial bottleneck that downsamples by 2, (4) convLSTM-btk-4: convLSTM with a spatial bottleneck that downsamples by 4. Using the deterministic case as the basis (train-test s-level=0), DNN quality is of the order $1 \approx 2 > 3 > 4$.

#### F.3.2 OTHER EVALUATION METRICS

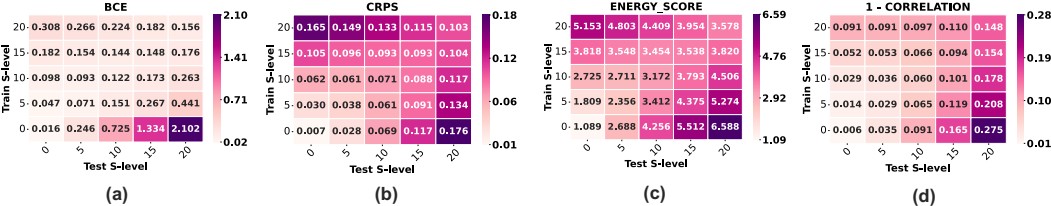

Figure 20: Matrix heatmap for error-based scores. First three, BCE, CRPS, Energy Score are proper scoring rules. Last one, is a popular spatial correlation based evaluation metric. None of them exhibit the sensitivity to stochasticity which can be observed for ECE.

### F.4 LONG HORIZON BEHAVIOR OF ECE VS. BASELINES

Figure 22 illustrates the DNN performance for the forest fire model using AUC-PR, MSE, and ECE when the s-levels of the training and test data match, corresponding to the diagonal in Figure

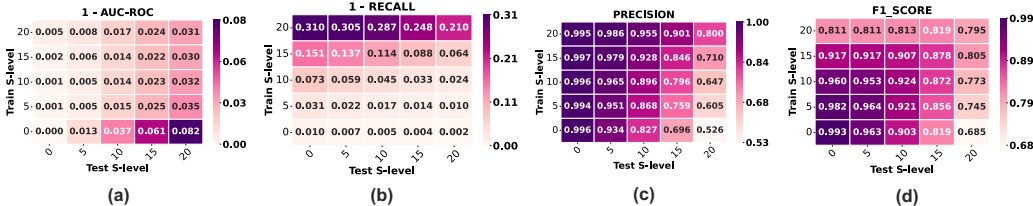

Figure 21: Matrix heatmap for other classification based metrics.

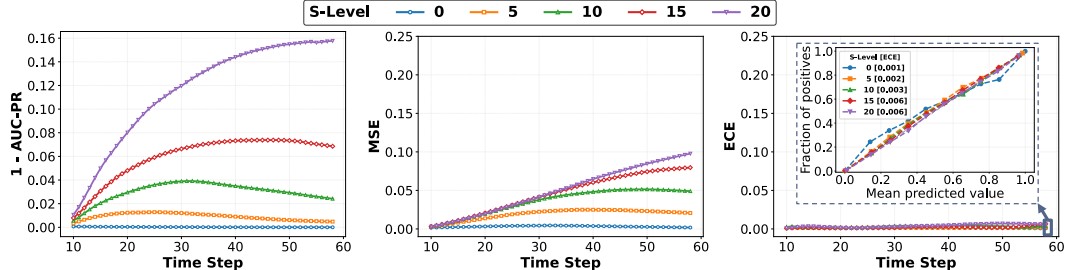

Figure 22: Long Horizon Performance of the DNN using (a) AUC-PR, (c) MSE, and (d) ECE [calibration curve in inset]. ECE demonstrates that the DNN's long horizon predictions remain stable.

3. Each evaluation score aggregates predictions and GT across 300 test simulations per time step. While baselines show declining scores with increasing S-Level and longer prediction horizons, ECE scores remain stable, indicating faithfulness to the Statistic-GT. In the special case of deterministic evolution (S-Level 0), all three evaluation metrics agree. For a perfect predictor, Refinement is 0 for deterministic evolution since $\text{frac}(B_m) = 0$ or 1 (§3.4.2), resulting in stable MSE. AUC-PR (F2R) is effective since there is only one possible realization. AUC-PR is measuring the ability of the DNN to generalize over different forest and fire seed configurations considering deterministic evolution.

## F.5 ECE CONVERGENCE VS. NUMBER OF SAMPLES

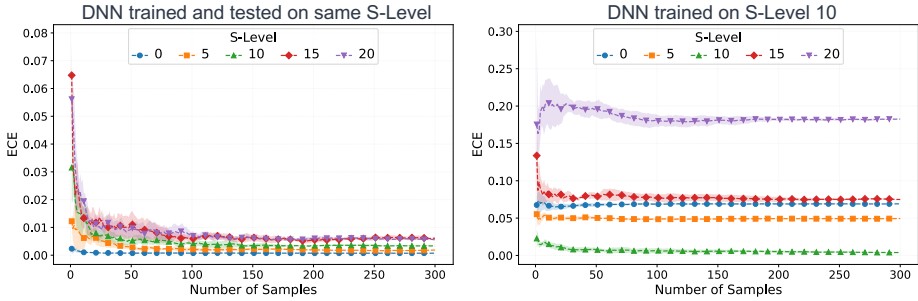

Figure 23: Testing the asymptotic guarantees of ECE. Each sample is a grid of predictions ($64 \times 64$) at t=59 in the test split of the synthetic forest dataset. As the number of samples increase in the calculation, ECE converges. We can observe, that for the asymptotic guarantees to kick in, sufficient number of samples are required.

# G  APPLICATION OF DEEP LEARNING IN WILDFIRE MODELING

## G.1  MODELS FOR WILDFIRE MODELING

Currently, forest fire spread models are categorized into three classes: empirical, semi-empirical, and physical. Empirical models, e.g., DNN modeling using remote sensing data (Jain et al., 2020), analyze fire data statistically without exploring combustion mechanisms. Semi-empirical models, often the preferred choice, like (Finney, 1998; Rothermel, 1972) integrate physical laws, such as heat transfer, but necessitate resource-intensive ground surveys for calculating model parameters (Finney, 1998). Physical models, involving complex equations for heat dynamics (Séro-Guillaume & Margerit, 2002), are too complex for broad application. All these classes of models are deterministic and do not explicitly assume fire dynamics to be stochastic.

Table 5: Current Evaluation Metrics

| Metric | Description |
|---|---|
| Precision | Proportion of true positive predictions among all positive predictions |
| Recall | Proportion of true positive predictions among all actual positive instances |
| Accuracy | Proportion of correct predictions among all predictions |
| F1-score | Harmonic mean of precision and recall |
| AUC-PR | Area under the precision-recall curve, evaluating trade-off between precision and recall |
| AUC-ROC | Area under receiver operating characteristic curve, evaluating trade-off between true positive rate and false positive rate |
| MSE | Mean squared error, measuring average squared difference between predicted and actual values |

Table 4: Selected works in Wildfire Prediction

| Work | Window Size | DNN | Evaluation Metric |
|---|---|---|---|
| (Radke et al., 2019) | Obs.: T Pred.: T+24h | CNN | F1-score, Recall, Accuracy |
| (Yang et al., 2021) | Obs.: [T-52w, T] Pred.: T+5w | CNN, LSTM | AUC-ROC, MSE |
| (Huot et al., 2022) | Obs.: T Pred.: T+24h | CNN | AUC-PR, Precision, Recall |

**DNN-based Modeling of Wildfires.** Wildfire prediction is a critical application area, driven by the increasing frequency and severity of wildfires (Westerling et al., 2006; Thompson et al., 2016). DNNs utilize multivariate observations—including weather, topology, and historical fire data—to predict future fire maps, identifying burn locations (Huot et al., 2022; Radke et al., 2019; Yang et al., 2021). These models learn the stochastic "rules" of fire evolution to improve prediction accuracy. Leveraging the broad spatial and temporal coverage of satellite (Huot et al., 2022) and aircraft-based sensor data (Doshi et al., 2019), DNNs capture fire dynamics with reduced operational costs compared to conventional tools like FARSITE (Finney, 1998), which rely on ground-based data.

The growing application of DNNs in wildfire prediction is evident in studies summarized in Table 4 and reviewed comprehensively in (Jain et al., 2020). For instance, FireCast (Radke et al., 2019) improves 24-hour wildfire perimeter prediction accuracy by 20% using satellite imagery (Finney, 1998). These models integrate covariates such as vegetation, terrain, and weather; for example, Huot et al. use a 64 x 64-pixel grid containing 11 observational variables (Huot et al., 2022).

DNN-based wildfire prediction operates in two phases: learning fire evolution rules from observations and applying this knowledge for future predictions. Models are trained using Binary Cross Entropy (BCE) loss and evaluated with metrics summarized in Table 5[2].

---

[2]AUC-ROC is not recommended due to class imbalance (Huot et al., 2022; Sofaer et al., 2019a).

## G.2 Observational variables in Next Day Wildfire Spread (NDWS) dataset

Table 6: Observational variables and their description in the NDWS dataset (Huot et al., 2022)

| Observational Variable | Description |
|---|---|
| Elevation | Terrain height above sea level |
| Wind Direction | The direction from which the wind originates |
| Wind Speed | Velocity of the wind |
| Minimum Temperature | Lowest daily temperature |
| Maximum Temperature | Highest daily temperature |
| Humidity | Amount of water vapor in the air |
| Precipitation | Amount of rain, snow, etc., that falls |
| Drought Index | Measure of dryness indicating drought conditions |
| Vegetation | Vegetation indices indicating plant health and coverage |
| Population Density | Number of individuals per unit area |
| Energy Release Component (ERC) | Indicator of fire potential energy release |

## G.3 Calibration Curve on NDWS dataset

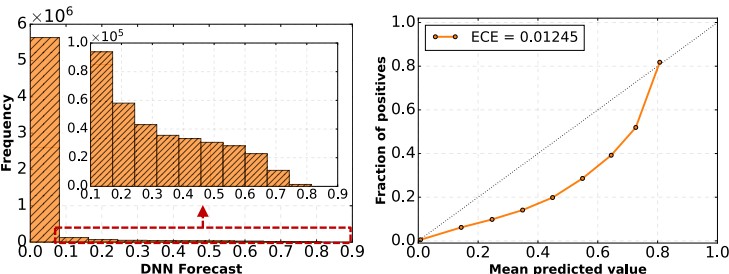

Figure 24: [Left] displays the histogram of forecasts generated by Conv-AE; [Right] shows a Calibration Curve that illustrates forecast accuracy in an interpretable manner. This curve suggests (1) the DNN's tendency towards overconfidence in mid-range forecasts, and (2) the accuracy of its probabilistic predictions is better for forecasts at the lower and upper ends of the probability spectrum.

## G.4 Resolving Rank Conflict in DNN Rankings

Table 7: Performance of DNNs on NDWS Dataset (ranks in parentheses by each evaluation metric)

| DNN | AUC-PR↑ | MSE↓ | ECE↓ |
|---|---|---|---|
| Conv-AE (Huot et al., 2022) | 0.2473 (5) | 0.0124 (4) | 0.0119 (4) |
| Conv-CA | 0.2631 (4) | 0.0146 (5) | 0.0207 (5) |
| AR-NCA (Kang et al., 2024) | 0.2790 (2) | 0.0099 (2) | **0.0012** (1) |
| SegFormer (Xie et al., 2021) | 0.2727 (3) | 0.0100 (3) | 0.0020 (2) |
| U-Net (Ronneberger et al., 2015) | **0.3302** (1) | **0.0096** (1) | 0.0023 (3) |

Conflicts in model rankings are key to model selection (Heaton et al., 2018). We simulate selecting the optimal DNN and explore how ECE complements existing metrics.

**DNN Models used.** We use five architectures: Conv-AE, a convolutional autoencoder from Huot et al. (Huot et al., 2022); Conv-CA, a modified Conv-AE without the spatial bottleneck (details in Appendix E.1); AR-NCA, an Attentive Recurrent Neural Cellular Automata for locally interacting discrete systems like forest fires (Kang et al., 2024); SegFormer, a transformer-based segmentation model (Xie et al., 2021); and U-Net, a convolution-based model (Ronneberger et al., 2015).

**Results and Discussion.** Table 7 presents evaluation scores (AUC-PR, MSE, and ECE) for five DNNs on the test split, revealing rank conflicts between the metrics. This raises the question: Which metric

best reflects model performance? Based on our findings, each metric evaluates a different system property: AUC-PR measures fidelity to the Observed-GT, ECE assesses fidelity to the Statistic-GT, and MSE lies somewhere in between. In complex systems, failure to replicate the Observed-GT isn't necessarily problematic due to inherent stochasticity, but a high ECE indicates a fundamental mismatch between the trained DNN and the test data, meaning the model should not be used. The evaluation strategy should prioritize ECE as the first metric, as a high ECE indicates the DNN has failed to capture the test data's stochasticity. Afterward, fidelity to realization metrics should be applied. This study ultimately advocates for a multi-faceted evaluation approach, with fidelity to the stochastic process as a crucial factor. While the NDWS dataset has a one-step forecast horizon, limiting our ability to assess ECE's stability over time, future work with datasets featuring longer horizons could explore this further.

# H  A PRACTICAL USER GUIDE FOR EVALUATING COMPLEX SYSTEMS

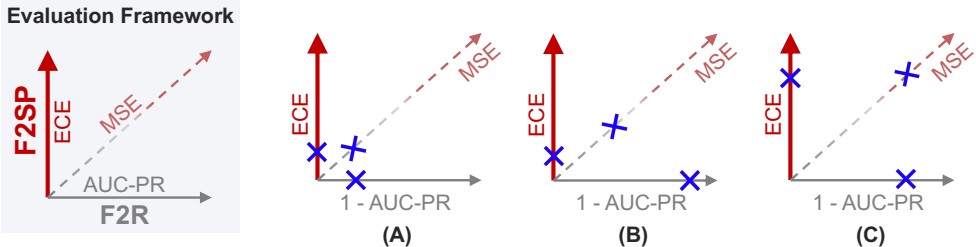

Figure 25: Visualization of prediction performance across different scenarios within the proposed evaluation framework. The evaluation framework distinguishes between two criteria: **F2R** and **F2SP**. **F2R** evaluates how closely the model predictions match a single observed outcome, assessed using metrics like AUC-PR (plotted along the x-axis as $1 - \text{AUC-PR}$, where lower values are better). **F2SP** evaluates whether the model captures the underlying stochastic process, measured by ECE (y-axis, where lower values indicate better calibration). The dashed gray line represents MSE, which balances F2R and F2SP. The $\times$ symbols represent the DNN's performance for different metrics. **(A)** Low errors across all metrics indicate the DNN matches the Observed-GT and captures the stochastic process. **(B)** High AUC-PR error but low ECE suggests the DNN has learned the stochastic process but fails to replicate the single Observed-GT. **(C)** High errors across all metrics suggest the DNN fails to match both the Observed-GT and the stochastic process. This visualization helps users analyze model performance across different evaluation criteria. See Table 8 for detailed interpretation.

Complex systems are modeled as a set of interacting components. These components exhibit random interactions, resulting in highly divergent outcomes and posing unique challenges for evaluation. This guide introduces a practical evaluation framework for machine learning (or statistical) models that forecast the behavior of such high-dimensional complex systems. The framework evaluates model predictions against ground truth while accounting for the inherent randomness in system evolution.

Traditional evaluation criteria focus on matching a single observed outcome. However, in stochastic systems, this outcome represents just one sample of the underlying process, as identical starting points can yield diverse results. This makes the current evaluation criteria inadequate.

Our evaluation framework offers a unique perspective on model performance in stochastic complex systems by distinguishing between **Fidelity to Realization (F2R)**, which focuses on matching a single observed outcome, and **Fidelity to Stochastic Process (F2SP)**, which evaluates whether a model captures the system's broader stochastic dynamics (§3). Using metrics like AUC-PR, ECE, and MSE, the framework provides a cohesive evaluation strategy by integrating these metrics within a unified structure. Rather than treating them independently and risking rank conflicts (§G.4), this framework aligns the metrics to offer a comprehensive and actionable understanding of a model's ability to balance individual outcome prediction with capturing the underlying stochastic process. Figure 25 visually summarizes this framework, highlighting different scenarios based on these metrics.

*Examples of Applications:*

- **Remote Sensing Grids:** Cells represent land areas, e.g., wildfire spread or weather events.
- **Virtual Grids:** Cells represent assets in financial markets or road segments in traffic systems.
- **Epidemiology:** Cells track infection in regions or neighborhoods.

The next page provides a detailed explanation of the framework's implementation and interpretation.

FRAMEWORK REQUIREMENTS

**Modeling Complex Systems on a Grid:** We model the system on a grid of size $H \times W$, where:

- **Grid Cells:** Each cell $(i, j)$ at time $t$ occupies one of $m$ states, $s_{t,(i,j)} \in \{s_1, \ldots, s_m\}$.
- **Target State:** A specific state $s^*$ is tracked over time for each grid cell.
- **Observed Ground Truth (Observed-GT):** For state $s^*$, define $b_{t,(i,j)} = 1$ if $s_{t,(i,j)} = s^*$, else 0. Collectively, these values form the ground truth grid $B_t = \{b_{t,(i,j)}\}^{H \times W}$.
- **DNN Predictions:** For each grid cell, the DNN predicts $\hat{p}_{t,(i,j)} \in [0, 1]$, representing the probability of $s^*$. These predictions form the grid-level output $\hat{P}_t = \{\hat{p}_{t,(i,j)}\}^{H \times W}$, which is compared against the corresponding $B_t$ over the forecast horizon $t = 1, 2, \ldots, T$.

CALCULATING EVALUATION METRICS

For each time step $t$, calculate evaluation metrics using ground truth $B_t$ and predictions $\hat{P}_t$ (standard library implementations in footnotes):

- **AUC-PR:** Measures how well predictions match the Observed-GT (*F2R*)[3].
- **ECE:** Assesses calibration against expected outcomes (*F2SP*)[4].
- **MSE:** Balances *F2R* and *F2SP*[5].

  **Note:** Ensure the grid size $H \times W$ is large enough for reliable metric computation. Larger grids or batch sizes improve reliability (see limitations of ECE in §7).

**Example Usage:** In our evaluation, we tested 300 simulations where the DNN observed the first 10 frames and predicted the next 50. This resulted in $300 \times 50 \times 64 \times 64$ ($B \times T \times H \times W$) ground truth samples ($B_t$) and predictions ($\hat{P}_t$), each of size $64 \times 64$. For the long-horizon plot in Figure 4, scores were calculated for each time step $T$ along the $B$ axis. For the stochastic matrix in Figure 3, these scores were averaged across time into a single value. Metrics can also be computed in batches for finer-grained analysis, provided a sufficiently large batch size is used for convergence (see §F.5).

INTERPRETING EVALUATION SCENARIOS WITH THE PROPOSED FRAMEWORK

Table 8 summarizes three evaluation scenarios to help users interpret results and their implications.

Table 8: Evaluation Scenarios and Their Impact.

| Scenario | $1 -$ **AUC-PR** | ECE | Interpretation and Impact |
|---|---|---|---|
| **A. Low Errors Across All Metrics** | Low | Low | The DNN matches the Observed-GT and is well-calibrated, capturing both the specific outcome and the underlying stochastic process. Ideal for low-stochasticity environments, this indicates the model is suitable for deployment. |
| **B. High AUC-PR Error but Low ECE** | High | Low | The DNN is well-calibrated but doesn't match the Observed-GT. It captures the stochastic process but fails to replicate a single realization, making it useful for high-stochasticity settings or longer prediction horizons. Predictions provide valuable insights into risk and variability. |
| **C. High Errors Across All Metrics** | High | High | The DNN is poorly calibrated and fails to match the Observed-GT, indicating fundamental issues with model training or system complexity. Predictions are unreliable and unsuitable for decision-making. |

---

[3]Scikit-learn AUC documentation
[4]Uncertainty-calibration documentation
[5]Scikit-learn MSE documentation

