# OpenReview forum: "Has the Deep Neural Network learned the Stochastic Process? An Evaluation Viewpoint"
_ICLR.cc/2025/Conference — ICLR 2025 Poster_

### Official Review · Reviewer_iuWX · 2024-10-23

**Soundness:** 2
**Presentation:** 3
**Contribution:** 2
**Rating:** 6
**Confidence:** 2

**Summary:**

This paper introduces a novel stochasticity-compatible evaluation strategy for assessing existing models in the context of complex systems. The author justifies the Expected Calibration Error (ECE) as suitable for assessing the model fidelity of stochastic systems through both simulation environments and real-world data.

**Strengths:**

1. Evaluating model fidelity on the stochastic system is significant and has wide applications.
2. The paper is well-motivated and both the dataset and experiments are thorough.

**Weaknesses:**

1. Although the author attempts to explain the difference between their work and ECE in deep learning in Lines 282-288, it appears to me this work is still a direct application of using ECE to evaluate the model performance on a stochastic system. The author is encouraged to discuss more in-depth about the distinction between ECE in the proposed method (stochasticity comes from evolving in the environment, aka, Statistic-GT) and ECE in previous works (stochasticity comes from the output distribution).
2. In Lines 243-244, the author claims that Statistic-GT is more stable than classification-based metrics, but I could not find any evidence related to calculating ECE on Statistic-GT is less sensitive to the system variance than MSE. Is there any theoretical support for using ECE over MSE on stochastic systems with different noise levels and could the author clarify it a bit more?

**Questions:**

I'm curious about how the author evaluates ECE at time $t$ based on Statistic-GT $P_{t}$. Do we have to simulate it again from $t=0$ for $N$ times or we can sample states from $t-1$ and go forward $N$ times (the system is Markov)? Can we still apply ECE on Statistic-GT when the system is not Markov?

---

> ### Author Response · Authors · 2024-11-21
> **Authors' Response (1/2)**
>
> > Although the author attempts to explain the difference between their work and ECE in deep learning in Lines 282-288, it appears to me this work is still a direct application of using ECE to evaluate the model performance on a stochastic system. The author is encouraged to discuss more in-depth about the distinction between ECE in the proposed method (stochasticity comes from evolving in the environment, aka, Statistic-GT) and ECE in previous works (stochasticity comes from the output distribution).
> >
>
> Thank you for raising this point. To clarify the distinction between the use of ECE in our work and in prior studies, we have revised the discussion in the manuscript (L293–301). Unlike prior works, where ECE is primarily used to measure DNN output calibration in static tasks (e.g., image or text classification), our study highlights its unique suitability for evaluating fidelity to a system property (Statistic-GT) in multivariate stochastic systems. We demonstrate that ECE serves not just as a tool for measuring output calibration but as a critical metric for testing F2SP, effectively addressing system randomness. Furthermore, we establish perfect calibration as a necessary condition for F2SP—a fundamental insight absent in prior studies. This reframing positions calibration as central to the evaluation of stochastic systems, rather than a secondary consideration. To provide additional context, the discussion in Appendix C.2 has been expanded to further support the main paper. We hope this enhanced discussion addresses your concern.
>
> > In Lines 243-244, the author claims that Statistic-GT is more stable than classification-based metrics, but I could not find any evidence related to calculating ECE on Statistic-GT is less sensitive to the system variance than MSE.
> >
>
> Thank you for raising this important point. To clarify, we do not claim that Statistic-GT is more stable than classification-based metrics. Rather, we claim that Statistic-GT is a more stable property than Observed-GT, as a given stochastic process has only one Statistic-GT but can have many Observed-GTs. To avoid confusion, we have rephrased the text in the manuscript to make this distinction clear (L254).
>
> To address your question, we have updated the discussion in Appendix F.1 (Figure 16) to include ECE in the plot analyzing the impact of macro-variance on metric sensitivity. The revised discussion (L1389–1394) highlights that both MSE and ECE exhibit lower sensitivity to macro-variance due to their inherent asymptotic convergence guarantees. However, their steady-state behavior differs: MSE's steady-state value is directly influenced by \(Var[Z_t]\), making it dependent on this variance, whereas ECE's steady-state value remains unaffected by \(Var[Z_t]\) upon convergence. This distinction, illustrated in the newly added Figure 17, underscores ECE's suitability for testing fidelity to Statistic-GT, as it remains unaffected by \(Var[Z_t]\), the system property that quantifies randomness.
>
> > Is there any theoretical support for using ECE over MSE on stochastic systems with different noise levels and could the author clarify it a bit more?
> >
> Regarding theoretical support for using ECE over MSE in stochastic systems with different noise levels: we formally and empirically demonstrate across various noise levels that ECE uniquely tests F2SP, while MSE does not. This distinction positions ECE as a critical metric for these systems. Figures 3 and 4 in the main paper provide empirical evidence, while Sections 3.4.1 and 3.4.2 offer theoretical insights into ECE's unique behavior compared to MSE in stochastic systems.

---

> > ### Author Response · Authors · 2024-11-21
> > **Authors' Response (2/2)**
> >
> > > I'm curious about how the author evaluates ECE at time  $t$ based on Statistic-GT $P_t$. Do we have to simulate it again from t = 0 for N  times or can we sample states from  t - 1  and go forward N  times (the system is Markov)? Can we still apply ECE on Statistic-GT when the system is not Markov?
> > >
> >
> > Thank you for your insightful question regarding the evaluation of ECE at time $t$ based on the Statistic-GT $P_t$. We appreciate the opportunity to clarify both the computation of Statistic-GT and its applicability across Markovian and non-Markovian systems.
> >
> > To compute Statistic-GT $P_t$, we require a distribution of possible system states at time $t$, generated through multiple realizations of the system's evolution. Two primary methods can achieve this: $1.$ simulating from $t = 0$ for $N$ realizations up to time $t$, or $2.$ starting from the state at $t - 1$ and simulating $N$ trajectories forward to $t$, leveraging the Markov property if applicable. In our work, we employ the first method—simulating from $t = 0$—to capture the full stochastic evolution of the system. This approach accounts for the accumulation of stochastic effects over time and the potential divergence of trajectories due to system sensitivity to initial conditions. While the second method can be used for Markovian systems, the first method ensures consistency and applicability across both Markovian and non-Markovian systems.
> >
> > Regarding non-Markovian systems, the computation of Statistic-GT does not rely on the Markov assumption. In such systems, where the future state depends on a sequence of past states rather than just the current state, multiple realizations from the same initial conditions still allow us to approximate $P_t$. This makes ECE applicable regardless of whether the system exhibits Markovian properties. Additionally, while Statistic-GT is used conceptually in our study to demonstrate ECE's fidelity to a system property, it is important to emphasize that ECE can be directly computed using the Observed-GT and DNN predictions. The core challenge addressed in our paper is showing that ECE effectively evaluates fidelity to Statistic-GT without requiring its explicit computation in real-world scenarios. The introduction of the paper has been revised to make this clear.

---

### Official Review · Reviewer_HXsj · 2024-10-30

**Soundness:** 3
**Presentation:** 3
**Contribution:** 3
**Rating:** 8
**Confidence:** 5

**Summary:**

This work offers a new perspective on evaluating DNNs in stochastic complex systems by emphasizing  the importance of capturing underlying the stochastic process. Traditional evaluation methods assess the DNN’s ability to replicate the observed ground truth but fail to measure the DNN’s learning of the underlying stochastic process. This paper proposes a new property called Fidelity to Stochastic Process, representing the DNN’s ability to predict the ground truth of the stochastic process, and introduces an evaluation metric that exclusively assesses fidelity to  the ground truth of the stochastic process. The Expected Calibration Error is used to evaluate the fidelity to ground truth of statistic process. Empirical experiments on synthetic datasets (including wildfire, host-pathogen, and stock market models) and real-world wildfire data are used to show the measurement of fidelity to stochastic process by Expected Calibration Error.

**Strengths:**

The paper offers a new perspective on evaluating DNNs by considering DNNs as stochastic processes and uses a widely used criteria in Bayesian Deep Learning application to assess the fidelity to stochastic process. This work clearly explains the Expected Calibration Error is used to assess DNN modes in three synthetic cases and one real world case.

**Weaknesses:**

This paper is well organized and well written, several minor issues should be addressed: (1) The explaination of figures is not sufficient, e.g., in Figure 2 (1), the label for x-axis is not specified (I guess it is time?), either add a label or explain it in the captions. Same problems also exist in Figure 4. (2) This work examines ECE on three synthetic environments (forest fire, host-pathogen and stock market models) and a real world wildfire spread dataset. I can tell that these datasets are all multivariate either for classification or regression. Maybe due to the limit of pages, the authors didn't include the experiments on images. I suggest the authors add some discussions or comments in the paper.

**Questions:**

As mentioned in the "Weaknesses" part.

---

> ### Author Response · Authors · 2024-11-21
> **Authors' Response**
>
> > The explaination of figures is not sufficient, e.g., in Figure 2 (1), the label for x-axis is not specified (I guess it is time?), either add a label or explain it in the captions. Same problems also exist in Figure 4.
> >
>
> Thank you for the feedback. We have updated the figures and captions to address these issues. For Figure 2, we added "time" as the x-axis label. For Figure 3 (previously Figure 4), we clarified the axes in the caption (L373): *"The x-axis (test S-Level) and y-axis (train S-Level) are consistent across all matrices."*
>
> > This work examines ECE on three synthetic environments (forest fire, host-pathogen and stock market models) and a real world wildfire spread dataset. I can tell that these datasets are all multivariate either for classification or regression. Maybe due to the limit of pages, the authors didn't include the experiments on images. I suggest the authors add some discussions or comments in the paper.
> >
>
> Thank you for your insightful comment. You are correct that this work focuses on multivariate predictions in complex systems and does not include experiments on univariate tasks or image-based scenarios. For univariate cases, such as image classification or object detection, the traditional context is not forecasting, which differs from our focus. Additionally, ECE requires a sufficient number of samples for reliable calibration error estimates (see L523). In our framework, grid-level predictions contribute significantly to ECE's binning process, which would not occur in univariate settings, introducing additional challenges.
>
> For multivariate cases in vision, such as segmentation map forecasting or stochastic video prediction, there are conceptual parallels to our work. For example, predicting \(s*\) states across a grid is analogous to forecasting segmentation maps. Although such tasks are not traditionally classified as complex systems, their problem formulation and local pixel interactions in videos provide a compelling analogy. A more detailed exploration of these tasks would require adjustments to the current framework, which we suggest as an exciting direction for future work.
>
> To ensure clarity, we have expanded the related works section to highlight that vision tasks, such as segmentation map forecasting and stochastic video prediction, predominantly rely on F2R evaluations, opening up the possibility of applying F2SP evaluation strategies in this context (L500). Additionally, Appendix C.4 has been updated to reflect this discussion. In our future work section (Section 7), we propose exploring how F2SP could be extended to vision tasks (L531).

---

> > ### Author Response · Authors · 2024-11-23
> > **Clarification on Multivariate Focus and Potential Extensions for Univariate Tasks**
> >
> > We have updated the text to address your comment regarding this work's focus on multivariate tasks rather than univariate ones. Specifically, the discussion in the Limitations of ECE section (L518-526) has been refined to explicitly state that our work focuses on multivariate prediction tasks (L523). Additionally, we explain why the convergence of ECE poses challenges for univariate forecasting tasks due to the sample size requirements. To address this limitation, we propose a potential avenue for extending the framework by leveraging improved calibration error estimators that offer stronger convergence guarantees for smaller test sets (L525). The added text is highlighted in blue.
> >
> > The reviewer's comments helped us directly identify two key avenues for future work: extending the framework to univariate tasks and applying it to vision-based problems. While we had included related discussions in the paper, the feedback allowed us to identify these opportunities and propose these directions more explicitly. Thank you for highlighting these important points.

---

> ### Comment · Reviewer_HXsj · 2024-11-28
>
> I have read the authors' response and agree with them, therefore I updated my score. Thank you!

---

> > ### Author Response · Authors · 2024-12-01
> > **Thank you for the feedback**
> >
> > Dear Reviewer,
> >
> > Thank you for your feedback and for taking the time to review our response. We also appreciate you updating the score for the contribution. However, we noticed that the overall score did not change and wanted to kindly check if this was intentional or perhaps an oversight.
> >
> > We are grateful for your engagement and the opportunity to address your concerns. Please let us know if there’s anything further we can clarify.
> >
> > Best regards,
> > The Authors

---

### Official Review · Reviewer_sNm1 · 2024-10-31

**Soundness:** 3
**Presentation:** 1
**Contribution:** 2
**Rating:** 6
**Confidence:** 2

**Summary:**

This paper presents a study on evaluating deep neural networks designed to forecast the evolution of stochastic complex systems. The authors identify a gap in traditional evaluation methods—such as threshold-based classification metrics and error-based scoring rules—which focus on a model's ability to replicate observed ground truth but fail to assess how well the model has learned the underlying stochastic process. To address this issue, they introduce a new property called Fidelity to Stochastic Process, representing the DNN's ability to predict the statistical ground truth of the stochastic process.

The paper proposes using the Expected Calibration Error (ECE) as an evaluation metric that satisfies the necessary conditions for assessing fidelity to statistical ground truth. This work underscores the importance of capturing the underlying stochastic processes in deep neural networks  evaluations for complex systems.

**Strengths:**

The paper makes a significant contribution by introducing the concept of Fidelity to Stochastic Process (F2SP), a novel evaluation criterion specifically designed to assess a DNN's ability to learn the underlying stochastic interactions in complex systems.

The authors provide a rigorous formalization of F2SP within a stochastic framework, establishing clear criteria for its valid measurement. The use of Expected Calibration Error (ECE) as an evaluation metric is well-justified.

**Weaknesses:**

I found it hard to read the paper because there was a lack of consistency in the acronyms, the authors would redefine them in several parts of the text again and again. I addressed my comments on text in the questions section.

In the tables, the best neural networks based on each criterion are not highlighted, which makes it difficult to the reader to infer and correlate the arguments in the text. I addressed my comments on text in the questions section.

The focus of the paper is primarily on binary or discrete prediction tasks, leaving out regression tasks where the definition of calibration is more complex. While the authors acknowledge this and suggest it as an area for future work, the current scope limits the immediate applicability of the findings to a broader range of problems involving continuous outcomes.

Additionally, the use of the NDWS dataset, which is restricted to next-day predictions, prevents the assessment of ECE over longer time horizons, which are common in many complex systems. Could you elaborate on how future work might address this limitation?

The paper highlights the lack of open-source complex system datasets as a barrier to broader validation. Are there any ongoing initiatives or plans to develop, collect, or standardize such datasets?

**Questions:**

L50: Is --> is (lowercase)
Fig1: no need to write the whole name, you can use acronyms because they're already defined in the text, however MSE is not defined at this point.
L88: fidelity to realization --> F2R (it was already defined previously, so you can use the acronym)
L99: the notation of the dimension of the real vector O_t is confusing, what is (R^n)^(H x W), is n = H x W? If so, make that explicit.
Table 1: some rows end with full stop, other don't. Please make it consistent. Either all with or all without.
I find it odd to place Figures in columns as Figure 1 (which has a large top white margin) and Figure 3. I would suggest column figures into one row figure with multiple subfigures as you did with Figure 2.
L201: Isn't the indicator variable already defined as B_t in L99? Why defining again with different notation?
L298: MSE already defined in text previously, no need to write the whole name again.
L516: ECE already defined in text previously, no need to write the whole name again.
Table 2 and Table 7: highlight the best performing DNNs.

---

> ### Author Response · Authors · 2024-11-21
> **Authors' Response (1/2)**
>
> > I found it hard to read the paper because there was a lack of consistency in the acronyms, the authors would redefine them in several parts of the text again and again. I addressed my comments on text in the questions section.
> L50: Is --> is (lowercase) Fig1: no need to write the whole name, you can use acronyms because they're already defined in the text, however MSE is not defined at this point. L88: fidelity to realization --> F2R (it was already defined previously, so you can use the acronym) L99: the notation of the dimension of the real vector O_t is confusing, what is (R^n)^(H x W), is n = H x W? If so, make that explicit. Table 1: some rows end with full stop, other don't. Please make it consistent. Either all with or all without.
> >
>
> Thank you for your detailed feedback. We have addressed all the mentioned issues: acronyms are now used consistently throughout the text without unnecessary redefinitions. The notation for \(O_t\) has been clarified (L120). Additionally, we standardized punctuation in Table 1. These updates ensure consistency and improve readability.
>
> > In the tables, the best neural networks based on each criterion are not highlighted, which makes it difficult to the reader to infer and correlate the arguments in the text. I addressed my comments on text in the questions section.
> Table 2 and Table 7: highlight the best performing DNNs.
> >
>
> Thank you for the suggestion. In Table 2, we added a color gradient to the evaluation metric columns to indicate relative performance, with red representing worse scores and lighter shades representing better scores. The caption was updated to reflect this change, improving readability. For Table 7, we highlighted the best-performing DNN, making the rank conflict more prominent and reinforcing the key theme of how to select the best model. These updates enhance clarity and help readers better correlate the text with the tables.
>
> > The focus of the paper is primarily on binary or discrete prediction tasks, leaving out regression tasks where the definition of calibration is more complex. While the authors acknowledge this and suggest it as an area for future work, the current scope limits the immediate applicability of the findings to a broader range of problems involving continuous outcomes.
> >
>
> Thank you for your thoughtful comment. While we acknowledge that the current scope focuses on binary and discrete prediction tasks, these tasks span a wide range of applications in complex systems, as highlighted in the references. This focus enables us to rigorously develop a foundational framework for assessing fidelity to system properties like the Statistic-GT without diluting the work’s impact across disparate problem classes.
>
> We agree that extending these findings to regression tasks with continuous outcomes is an important area for future work. However, expanding the scope within this paper, which already spans more than 30 pages including the appendix, would risk overwhelming readers and obscuring the primary contributions. We believe this work provides a robust foundation for future extensions and appreciate your perspective.
>
> > Additionally, the use of the NDWS dataset, which is restricted to next-day predictions, prevents the assessment of ECE over longer time horizons, which are common in many complex systems. Could you elaborate on how future work might address this limitation?
> >
>
> Thank you for highlighting this limitation. We acknowledge that the NDWS dataset, restricted to next-day predictions, limits the ability to robustly validate ECE over longer time horizons. Addressing this limitation would require datasets that span multiple time horizons (L539), many of which already exist but are not open-sourced. Future work could leverage such datasets to further validate our findings.
>
> That said, our evaluation framework remains broadly applicable, even for next-timestep predictions. Importantly, if the time horizon between consecutive steps is large (e.g., monthly versus daily predictions), the relative importance of F2SP versus F2R shifts. For shorter horizons, F2R may be more critical, whereas for longer horizons, F2SP becomes more relevant. This distinction, along with guidance on interpreting results in these contexts, is discussed in the practical user guide added to the Appendix H.
>
> > The paper highlights the lack of open-source complex system datasets as a barrier to broader validation. Are there any ongoing initiatives or plans to develop, collect, or standardize such datasets?
> >
>
> We are exploring collaborations with domain experts, such as those in forest fire modeling, to assess the feasibility of assembling standardized datasets. We acknowledge the inherent challenges in unifying diverse data sources due to varying system constraints. We hope this work underscores the importance of open-source datasets and inspires broader efforts within the research community to develop and standardize such resources.

---

> > ### Author Response · Authors · 2024-11-21
> > **Authors' Response (2/2)**
> >
> > > I find it odd to place Figures in columns as Figure 1 (which has a large top white margin) and Figure 3. I would suggest column figures into one row figure with multiple subfigures as you did with Figure 2.
> > >
> >
> > Thank you for pointing this out. We agree that placing Figures 1 and 3 in columns with large top white margins was not optimal. In response, we have merged these two figures into Figure 1. We did the same for Figure 5 and Table 2. This adjustment improves the visual consistency of the paper and makes better use of space. We appreciate your suggestion, as it enhances the overall presentation of our work.
> >
> > > L201: Isn't the indicator variable already defined as B_t in L99? Why defining again with different notation? L298: MSE already defined in text previously, no need to write the whole name again. L516: ECE already defined in text previously, no need to write the whole name again.
> > >
> >
> > Thank you for the feedback. We have addressed these issues by ensuring consistent use of previously defined notations and abbreviations. The redundant redefinitions of \(B_t\), MSE, and ECE have been removed. Additionally, the definition of the micro random variable has been rephrased to reference \(B_t\) directly for clarity (see L226).

---

> > > ### Comment · Area_Chair_VRRs · 2024-11-26
> > >
> > > Dear reviewer,
> > >
> > > Please make to sure to read, at least acknowledge, and possibly further discuss the authors' responses to your comments. Update or maintain your score as you see fit.
> > >
> > > The AC.

---

> ### Author Response · Authors · 2024-12-01
> **Thank you for your feedback**
>
> Dear Reviewer,
>
> Thank you for your feedback and for increasing the score. We appreciate your insights and the time you’ve taken to help improve the paper.
>
> Best regards,
> The Authors

---

### Official Review · Reviewer_7Rfe · 2024-11-01

**Soundness:** 2
**Presentation:** 2
**Contribution:** 2
**Rating:** 6
**Confidence:** 3

**Summary:**

This paper presents a study evaluating deep neural networks (DNNs) within stochastic complex systems, emphasizing the importance of Expected Calibration Error (ECE) in measuring fidelity to stochastic processes. The findings are validated through multiple experiments and comparisons.

**Strengths:**

The topic of evaluating DNNs within stochastic complex systems is both intriguing and important.

In the primary evaluations, the author conducted experiments across various settings, including different DNN architectures, comparisons with multiple evaluation metrics, and diverse simulation tasks.

The main text clearly explains the difference between ECE in classical assessment and stochastic process settings.

**Weaknesses:**

The paper is somewhat difficult to follow. For example, providing a brief introduction to the structure of each section would enhance clarity, particularly in Sections 2 and 3. Additionally, it is difficult to grasp the main messages conveyed by the table in Figure 2(b). Furthermore, in lines 229–240, the macro-level concept is introduced abruptly, which may disrupt the clarity and readability of the main text.

The main findings' practical applicability appears limited. In real-world scenarios, data generally provides only a single observed outcome centered on observable ground truth (line 117). Since the primary evaluation is simulation-based, the controlled stochasticity falls short of capturing real-world complexity. The Statistic-GT is basically derived by normalizing the frequency of target state occurrences across multiple Monte Carlo simulations.

Minors:

M1. The original text for the abbreviation RV is not given.

M2. In Table 1, what about the possibility of recovery in the Host-Pathogen problem?

M3. In line 152, maybe consider using an alternative symbol for Moore neighborhood, instead of $\mathcal{N}$ (normally representing Gaussian distribution).

**Questions:**

A question arises regarding Figure 1: Can ECE be an effective metric for measuring F2R compared to other available metrics? Figure 1 suggests that the answer may be *no*.

An important indicator that ECE is a reliable measure is its diagonal pattern, showing low scores only when training and test S-Levels align, as illustrated in Figure 4. Could the authors provide theoretical insights to support this indicator?

---

> ### Author Response · Authors · 2024-11-21
> **Authors' Response (1/2)**
>
> > The paper is somewhat difficult to follow. For example, providing a brief introduction to the structure of each section would enhance clarity, particularly in Sections 2 and 3.
> >
>
> Thank you for the helpful feedback. To improve the paper’s clarity, we have added brief summary text at the beginning of Section 2 (L110) and Section 3 (L185) to provide readers with a clear overview of each section’s structure and objectives. Additionally, we made several updates throughout the paper aimed at enhancing overall clarity. These changes include revising figures, captions, and corresponding text to better align with the content and improve readability. All changes are highlighted in red for ease of review.
>
> > Additionally, it is difficult to grasp the main messages conveyed by the table in Figure 2(b).
> >
>
> Thank you for the feedback. We have revised the table in Figure 2(b) to enhance clarity and convey its main message more effectively. Updates include a red dashed line separating deterministic and stochastic simulation styles, a legend clarifying symbols ("✓: Random ✖: Fixed"), and explicit \(=0\) and \(>0\) markers under S-Level to differentiate setups. The "Stochastic Process" label highlights its connection to ESP and stochastic evolution, while grouping "Forest Configuration," and "Fire Seed Location," under "Initial Conditions" improves organization. The caption has also been updated to clearly articulate the table’s message.
>
> > Furthermore, in lines 229–240 (However, these systems explore  ….), the macro-level concept is introduced abruptly, which may disrupt the clarity and readability of the main text.
> >
>
> Thank you for the feedback. In the revised text, we have refined the discussion to explicitly connect the aggregation of micro-level comparisons with the derivation of a macro-level evaluation score, framing it as a standard approach for summarizing DNN performance. This establishes a logical progression leading to the formal definition of the Macro Random Variable (\( Z_t \)) as a grid-level summary of system behavior, aligned with the calculation of the evaluation metric (see L241,249). These changes ensure the text flows more naturally and improves clarity.
>
> > The main findings' practical applicability appears limited. In real-world scenarios, data generally provides only a single observed outcome centered on observable ground truth (line 117). Since the primary evaluation is simulation-based, the controlled stochasticity falls short of capturing real-world complexity. The Statistic-GT is basically derived by normalizing the frequency of target state occurrences across multiple Monte Carlo simulations.
> >
>
> Thank you for the comment. We acknowledge that in real-world scenarios, only a single Observed-GT is available, making direct computation of the Statistic-GT infeasible. To address this, we clarify that Statistic-GT represents a conceptual system property capturing the expected behavior across all possible outcomes. While simulation-based experiments allow us to validate the framework by explicitly calculating the Statistic-GT, our key contribution is showing that F2SP can be tested using only the Observed-GT. This ensures our framework aligns with real-world constraints, with ECE reliably assessing DNN fidelity to the stochastic process without requiring multiple realizations.
>
> We have added explicit acknowledgment of this challenge in the introduction (L78, L265) and revised the text to emphasize that ECE tests F2SP using only the Observed-GT (L99, L268). These updates clarify the practical applicability of our approach.
>
> > Minors:
> >
> >
> > M1. The original text for the abbreviation RV is not given.
> >
> > M2. In Table 1, what about the possibility of recovery in the Host-Pathogen problem?
> >
> > M3. In line 152, maybe consider using an alternative symbol for Moore neighborhood, instead of  (normally representing Gaussian distribution).
> >
>
> Thank you for highlighting these minor issues. We have addressed them as follows: the abbreviation *RV* has been explicitly defined, a note on the possibility of recovery in the Host-Pathogen problem has been added to Table 1, and the symbol for the Moore neighborhood in line 152 has been replaced to avoid confusion with Gaussian distribution notation.

---

> > ### Author Response · Authors · 2024-11-21
> > **Authors' Response (2/2)**
> >
> > > A question arises regarding Figure 1: Can ECE be an effective metric for measuring F2R compared to other available metrics? Figure 1 suggests that the answer may be *no*.
> > >
> >
> > Thank you for raising this question. We acknowledge that ECE, while effective for testing F2SP, is less suited for F2R evaluation due to its lack of a refinement term, which is crucial for capturing prediction sharpness and discriminative capabilities. We have updated the limitations section (L518–522) to incorporate this distinction, clarifying the different roles of ECE and classification-based metrics as you have pointed out.
> >
> > > An important indicator that ECE is a reliable measure is its diagonal pattern, showing low scores only when training and test S-Levels align, as illustrated in Figure 4. Could the authors provide theoretical insights to support this indicator?
> > >
> >
> > Thank you for the observation. Section 3.4.1 and 3.4.2 provides theoretical insights explaining the diagonal behavior of ECE, demonstrating why low scores occur when training and test S-Levels align. To enhance clarity, we have updated the caption of Figure 4 to explicitly reference the sections providing the theoretical insights, ensuring the connection is clear.

---

### Official Review · Reviewer_cBqL · 2024-11-08

**Soundness:** 4
**Presentation:** 4
**Contribution:** 4
**Rating:** 8
**Confidence:** 4

**Summary:**

the metric of expected calibration error is introduced and studied as a way to capture fidelity of a learned representation to an underlying stochastic process (rather than a single realization of that process, as with typical metrics like AUC or MSE).

**Strengths:**

Great paper, wonderfully practical and insightful; I've been looking for something like this for 5+ years! Nice eval on real-world data.
I started writing a thing I would like to you add and then discovered it was already in the paper (long horizon behaviour)

**Weaknesses:**

While overall the paper is very clear, some of the captions and explanations of the experiments/insights from them and how they tie to the figures could be improved.
Some specifics:
- first fig should say what you mean by realization, and F2R and F2SP should be bolded (not ital) to make them easy to find in the text. Observed GT should be explained a bit more, or maybe it would be enough to move the sentence currently after F2R to be the second sentence of the paragraph.
 - Fig 5 is unclear to me. What is the data, what is S-level, why is it "good" that the 20 vs 10 lines are far apart? All of this should be clear from the caption
 - the clarity wanes a bit as the paper goes on, and it's a bit confusing that you call it ECE vs. F2SP vs Statistic-GP. Do these different namings really serve something? It could be a lot more clear if you just have one naming.

**Questions:**

- I don't understand the second part of the critical question, "is it encountering different stochastic behaviours" (different from what)? how is the "differentness" relevant?
- While it's pretty clear to me how to use this immediately in my work, I think anyone who wasn't already aware they wanted exactly this might struggle. Could you provide something like a "practical users guide" for non-domain experts?
 - if the clarity of the plots can be improved, the naming of the stat/metric you're introducing, and improve it's "usability" to the community, I would be happy to upgrade my score. You've done great work and this would bring the paper to the level it deserves.

---

> ### Author Response · Authors · 2024-11-21
> **Author's Response (1/2)**
>
> > first fig should say what you mean by realization, and F2R and F2SP should be bolded (not ital) to make them easy to find in the text. Observed GT should be explained a bit more, or maybe it would be enough to move the sentence currently after F2R to be the second sentence of the paragraph.
> >
>
> Thank you for the valuable feedback. We have restructured and merged the original Figure 1 with the previous Figure 3 to contextualize F2R and F2SP in terms of Observed GT and Statistic GT. The updated Figure 1 provides a comprehensive overview of the paper, accompanied by a self-explanatory caption that clearly outlines the system property of interest, the evaluation criteria, and the corresponding metrics used to assess these criteria.
>
> In response to the comment about the text, we have revised the second, third, and fourth paragraphs of the introduction to address the suggestions provided by all reviewers. These changes aim to enhance the clarity of the arguments and resolve any confusion highlighted in the reviews. Please see the text highlighted in red in the revised pdf.
>
> > the clarity wanes a bit as the paper goes on, and it's a bit confusing that you call it ECE vs. F2SP vs Statistic-GP. Do these different namings really serve something? It could be a lot more clear if you just have one naming.
> >
>
> Thank you for highlighting this point. To address the concern about the clarity of naming conventions, we have explicitly defined the rationale behind the terms *ECE*, *F2SP*, and *Statistic-GT* in the revised text. Additionally, we have included a table in the updated Figure 1.b to further clarify the distinctions between these terms.
>
> Specifically:
>
> - *Statistic-GT* represents the *system property* we aim to evaluate fidelity to. It highlights the difference from *Observed-GT*, which refers to the observable outcomes of the system’s evolution.
> - *F2SP* is the *evaluation criterion* that tests fidelity to *Statistic-GT*, contrasting with *F2R*, which evaluates fidelity to *Observed-GT*.
> - *ECE* is an existing *evaluation metric* that serves as a tool to assess *F2SP*.
>
> We have revised the introduction (Lines 43, 44, 50, and 77) and Figure 1.b to make these distinctions explicit. Moreover, we have ensured consistent use of *F2SP* and *F2R* throughout the paper as evaluation criteria, mentioning system properties only when relevant to the discussion. These changes aim to improve clarity and address the reviewer’s concerns directly.
>
> > - Fig 5 is unclear to me. What is the data, what is S-level, why is it "good" that the 20 vs 10 lines are far apart? All of this should be clear from the caption
> >
>
> Thank you for the feedback. We have updated the caption of Figure 5 (L403) to address your comments and ensure greater clarity. The revised caption now explicitly describes the data, the meaning of S-Level, and why the separation of the 20 vs. 10 lines is significant. For your convenience, the updated caption is provided below:
>
> *"Two DNNs were trained on 700 forest fire simulations with different S-Levels—10 (orange, low stochasticity) and 20 (blue, high stochasticity)—and evaluated on 300 test simulations with S-Level 20. Evaluation metrics (a) AUC-PR, (b) MSE, and (c) ECE were measured over an extended prediction horizon. AUC-PR shows similar trends for both models, failing to distinguish the stochastic mismatch, while MSE declines more steeply for the mismatch case but also shows a declining trend for both DNNs due to misalignment with the Observed-GT. ECE remains low and stable for the DNN trained on S-Level 20. This highlights ECE’s unique ability to track alignment with the Statistic-GT, unlike AUC-PR and MSE, which focus on the Observed-GT."*
>
> > I don't understand the second part of the critical question, "is it encountering different stochastic behaviours" (different from what)? how is the "differentness" relevant?
> >
>
> Thank you for pointing out the need for clarification. We have revised the text to make the critical question clearer and to emphasize the relevance of the "differentness." The updated phrasing (L45 onwards) is as follows:
>
> *"This focus on F2R raises a critical question when a DNN fails to match the Observed-GT: is the mismatch due to inherent stochastic variability, or does it result from exposure to a fundamentally different stochastic process that the DNN has not modeled? Understanding this distinction is crucial: a DNN that accurately captures the stochastic process, even if it mismatches the Observed-GT, may still offer valuable insights, whereas a failure to model the process entirely undermines its utility."*
>
> This revision explicitly clarifies the meaning of "different stochastic behaviors" and highlights the practical importance of the difference between failing to match the Observed-GT vs. failing to match the underlying stochastic process.

---

> > ### Author Response · Authors · 2024-11-21
> > **Author's Response (2/2)**
> >
> > > While it's pretty clear to me how to use this immediately in my work, I think anyone who wasn't already aware they wanted exactly this might struggle. Could you provide something like a "practical users guide" for non-domain experts?
> > >
> >
> > Thank you for the suggestion. We have added a practical user guide in Appendix H, which is now cross-referenced in the caption of Figure 1. This addition provides actionable steps for non-domain experts to apply the evaluation framework effectively and enhances the usability of our work.
> >
> > To further integrate this guide into the paper, we revised the Introduction (L104) to reflect the discussion on its practical implications. Additionally, we updated the text (L487 onward) to introduce and motivate the cohesive evaluation framework, explicitly linking it to the practical guide. This framework clarifies metric prioritization by visually organizing AUC-PR, ECE, and MSE to reflect their complementary roles in evaluating model fidelity to stochastic dynamics (F2SP) and specific outcomes (F2R).
> >
> > We believe these changes address the reviewer’s concern by providing clear guidance for broader usability and ensuring the framework’s practical relevance is well-articulated.
> >
> > > if the clarity of the plots can be improved, the naming of the stat/metric you're introducing, and improve it's "usability" to the community, I would be happy to upgrade my score. You've done great work and this would bring the paper to the level it deserves.
> > >
> >
> > Thank you for the encouraging feedback and for recognizing the contributions of our work. In response to your suggestions, we have made significant updates to improve the clarity of the plots, refine the naming conventions of the statistics and metrics, and enhance the usability of the framework for the broader community. We hope these revisions meet your expectations and elevate the paper to the level you envision. We would greatly appreciate your reconsideration of the score in light of these improvements.

---

### Author Response · Authors · 2024-11-21
**Summary of Revisions**

We thank the reviewers for their valuable feedback, which has significantly improved our paper. Peer review has been instrumental in clarifying and enhancing our findings, and we deeply appreciate the time and effort reviewers invested.

The reviewers acknowledged the paper's contribution to evaluating DNNs in stochastic complex systems, specifically the introduction of Fidelity to Stochastic Process (F2SP) and its rigorous formalization. They also noted the thorough experiments, the clear differentiation between classical and stochastic evaluation approaches, and the practical applicability demonstrated through real-world datasets.

We have revised the text in the paper, and the specific segments addressing reviewers’ comments have been highlighted in red. Key concerns and changes are summarized below. Revisions have been made in both the main paper and the Appendix.

---

## **Changes Made in the Main Paper**

### 1. **Practical Applicability and Usability**

**Pain Points**:
Reviewers raised concerns about the clarity of the study’s practical applicability. There was also some confusion about whether Statistic-GT is required to measure ECE, raising questions about ECE’s applicability for testing F2SP in practical scenarios.

**Our Response**:

- **Framework Visualization**: Redesigned Figure 1 into a cohesive evaluation framework that visually organizes metrics (AUC-PR, ECE, MSE), consolidating our insights.
- **Practical User Guide**: Added a user guide in **Appendix H** with actionable steps for applying and interpreting the evaluation framework. This guide is cross-referenced in the main text and figures for easy navigation.
- **Clarified Applicability**: Highlighted in the introduction and discussion sections that ECE can test F2SP using only the Observed-GT, demonstrating its practical applicability. To clarify, the concept of Statistic-GT was formalized in the paper to substantiate this claim.

---

### 2. **Figures and Tables**

**Pain Points**:
Figures and tables lacked clarity, with insufficient captions.

**Our Response**:

- Added missing axis labels and expanded captions to ensure all figures are self-contained, understandable, and clearly convey their main messages, making it easier to correlate them with the text.
- Merged the original Figures 1 and 3 into a **new Figure 1**, which provides a clearer visual overview of the evaluation framework, including:
  - The two system properties, Observed-GT and Statistic-GT.
  - The two evaluation criteria, F2R (Fidelity to Realization) and F2SP (Fidelity to the Stochastic Process).
  - The evaluation metrics that measure these criteria.
- Updated Figure 2.b to clarify its message by highlighting the different sources of randomness in the synthetic datasets used in our study.
- Added color gradients to Table 2 to illustrate score transitions from good to bad and improve alignment with the text.

---

### 3. **Theoretical Justification**

**Pain Points**:
Need for theoretical support differentiating ECE and MSE with respect to sensitivity to system variance, explaining ECE’s behavior and its preference over MSE in certain contexts.

**Our Response**:

- **Sensitivity Analysis**: Included ECE in the metric sensitivity analysis to macro-variance in **Appendix F.1**, demonstrating that while ECE and MSE exhibit similar sensitivity, their convergence asymptotes differ. Added Figure 17 to show that ECE's convergence behavior makes it better suited for testing fidelity to Statistic-GT compared to MSE. Revised discussions in Section 3.3 to cohesively connect these findings.
- **Referenced Supporting Sections**: Updated figure captions and main text to directly reference sections providing theoretical insights, ensuring readers can easily locate and understand the explanations.

---

### 4. **Scope and Limitations**

**Pain Points**:
Limited discussion on extending the framework to image-based tasks.

**Our Response**:
Expanded and revised the related works section (Section 6), **Appendix C.4**, and the future works section (Section 7) to discuss potential applications to vision tasks, such as segmentation map forecasting and stochastic video prediction.

---

### 5. **Terminology and Notation Clarification**

**Pain Points**:
Confusion around the terms ECE (Expected Calibration Error), F2SP, Statistic-GT, and their relationships.

**Our Response**:

- Provided clear and explicit definitions of all key terms in the introduction.
- Included a clarifying table in the updated Figure 1 to illustrate the distinctions between terms.

---

### 6. **Clarity, Readability, and Minor Corrections**

**Pain Points**:
Minor issues such as undefined abbreviations, inconsistent punctuation, notation errors, and formatting inconsistencies were also noted.

**Our Response**:
We conducted a thorough sweep of the paper to address all these issues. Details of the revisions are provided in the individual responses.

---

> ### Author Response · Authors · 2024-11-21
> **Summary of Changes made in the Appendix**
>
> - **Section C.2**: Expanded the discussion on the general use of ECE as an evaluation metric compared to its application in our work. Clarified how our approach broadens the utility of ECE in evaluating stochastic systems.
> - **Section C.4**: Added two new references on computer vision tasks related to predicting the evolution of segmentation maps. Reorganized the discussion to emphasize how the current evaluation strategies in the computer vision community are predominantly focused on F2R.
> - **Section F.1.2**: Updated Figure 16 to include ECE alongside other metrics, enabling a comparative analysis of their standard deviation versus \( Var[Z_t] \). Added a revised discussion explaining that both MSE and ECE show reduced sensitivity to macro-variance, but their convergence behaviors differ. Added Figure 17 to illustrate this distinction: MSE penalizes stochasticity effects, whereas ECE does not.
> - **Section H**: Added a practical user guide for evaluating DNN models in complex systems, providing actionable steps for applying the proposed framework.

---

> > ### Author Response · Authors · 2024-11-23
> > **Scope Clarification and Future Directions**
> >
> > In response to the **HXsj**’s insightful feedback, we made a small change to the paper (L523,526) to further clarify its scope and propose future extensions. Specifically, we updated the Limitations of ECE section to explicitly state that the work focuses on multivariate tasks and to explain why univariate forecasting poses challenges due to ECE's sample size requirements. Additionally, we proposed using improved calibration error estimators as a potential extension to address this limitation.

---

> > > ### Author Response · Authors · 2024-11-27
> > > **Final updates to the PDF before the submission deadline**
> > >
> > > We have updated the PDF with additional minor refinements based on reviewer feedback. Specifically, in response to Reviewer 7Rfe's feedback, we highlighted the broad applicability of our findings in the introduction, supported by evidence from the analysis of real-world data. This is reflected in the updated last point in the contributions list in the introduction (L99–102):
> > >
> > > > Beyond synthetic systems, we analyze real-world wildfire data, identifying instances where stochasticity disrupts traditional metrics and observing trends that align with our synthetic findings, reinforcing the practical applicability of our study.
> > >
> > > Additionally, we made minor cosmetic improvements to Figure 3 for better quality. The captions of Figures 3 and 4 were slightly refined to make their key messages clearer, incorporating feedback from all reviewers.

---

### Comment · Area_Chair_VRRs · 2024-11-26

Dear all,

The deadline for the authors-reviewers phase is approaching (December 2).

@For reviewers, please read, acknowledge and possibly further discuss the authors' responses to your comments. While decisions do not need to be made at this stage, please make sure to reevaluate your score in light of the authors' responses and of the discussion.

- You can increase your score if you feel that the authors have addressed your concerns and the paper is now stronger.
- You can decrease your score if you have new concerns that have not been addressed by the authors.
- You can keep your score if you feel that the authors have not addressed your concerns or that remaining concerns are critical.

Importantly, you are not expected to update your score. Nevertheless, to reach fair and informed decisions, you should make sure that your score reflects the quality of the paper as you see it now. Your review (either positive or negative) should be based on factual arguments rather than opinions. In particular, if the authors have successfully answered most of your initial concerns, your score should reflect this, as it otherwise means that your initial score was not entirely grounded by the arguments you provided in your review. Ponder whether the paper makes valuable scientific contributions from which the ICLR community could benefit, over subjective preferences or unreasonable expectations.

@For authors, please respond to remaining concerns and questions raised by the reviewers. Make sure to provide short and clear answers. If needed, you can also update the PDF of the paper to reflect changes in the text. Please note however that reviewers are not expected to re-review the paper, so your response should ideally be self-contained.

The AC.

---

### Meta-Review · Area_Chair_VRRs · 2024-12-20

**Metareview:**

The reviewers unanimously recommend acceptance (8-6-6-8-6). The paper presents a significant contribution for the evaluation of neural networks designed to forecast the evolution of stochastic complex systems. Reviewers recognize the importance of the work and the quality of the results. The author-reviewer discussion has been constructive and has led to a number of improvements to the paper, in particular regarding its presentation. The reviewers have raised some concerns about the practical applicability of the findings (e.g., the focus on binary prediction tasks, the limited horizon in the experiments), but the authors have provided convincing arguments arguing for the relevance of their work nonetheless. No major concerns have been raised by the reviewers. For these reasons, I recommend acceptance. I encourage the authors to implement the changes discussed with the reviewers in the final version of the paper.

**Additional Comments On Reviewer Discussion:**

The author-reviewer discussion has been constructive and has led to a number of improvements to the paper, in particular regarding its presentation.

---

### Decision · Program_Chairs · 2025-01-22

Accept (Poster)